# NOSTRA: ENABLING ROBUST ROBOT IMITATION VIA MULTIMODAL LATENT IMAGINATION

## ABSTRACT

Similar to humans, robots benefit from multiple sensing modalities when performing complex manipulation tasks. Current behavior cloning (BC) policies typically fuse learned observation embeddings from multimodal inputs before decoding them into actions. This approach suffers from two key limitations: 1) it requires all modalities to be present and in-distribution at test time, otherwise corrupting the latent state and leading to fragile execution; and 2) naive fusion across all inputs hinders learning from large-scale heterogeneous datasets, where only a subset of modalities may be informative at different phases of a task. We introduce NOSTRA, a multimodal state-space model that learns a modular per-modality latent representation, enabling flexible action prediction with or without specific inputs. BC-NOSTRA improves robustness to unseen noise by using KL divergence between inferred and imagined multimodal latents as a noise measure, and by employing latent imagination to predict action trajectories over arbitrary horizons. On a suite of MuJoCo-based tasks, BC-NOSTRA fits expert demonstrations up to six input modalities (multi-view RGB, depth, and proprioception), achieving over 20% higher performance under noisy evaluation. Furthermore, NOSTRA adaptively down-weights non-informative inputs, facilitating effective co-training on large heterogeneous robotics datasets with $\mathcal{O}(10k)$ demonstrations spanning diverse tasks and visual conditions. Finally, we demonstrate real-world deployment, where BC-NOSTRA achieves up to a 40% performance gain under camera occlusions on multiple manipulation tasks.

## 1 INTRODUCTION

Humans rely on a rich variety of sensing modalities – vision, hearing, touch, pressure, temperature – to perform everyday tasks. Crucially, not all modalities are necessary at once: when one becomes unavailable (e.g., vision under occlusion), we draw on experience and dead-reckoning to bridge the gap, and seamlessly return to the more informative signal once it becomes available again. Achieving human-level robustness in robotic object manipulation requires similar capabilities. Robots must be able to process diverse inputs such as multiple camera views, depth, and proprioception, each of which occupies a distinct subspace and often warrants its own encoder within a visuomotor policy. Equally important, they must learn to identify which inputs are useful at a given time and remain robust when some become noisy, unreliable, or entirely missing due to sensor failures. While state-of-the-art behavior cloning (BC) policies do incorporate multiple input modalities, they typically fuse their learned representations naively to predict actions (Mandlekar et al., 2021; Chi et al., 2023), a strategy that falls short of these requirements.

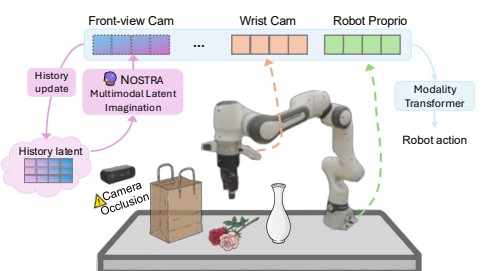

Figure 1: NOSTRA uses latent imagination to handle unforeseen noise in a multimodal robotic system. This allows it to maintain performance and robustness even with noisy or corrupt inputs.

Incorporating multimodal observations into visuomotor policies presents significant challenges, particularly in deciding when and how each modality should contribute to decision-making. Not

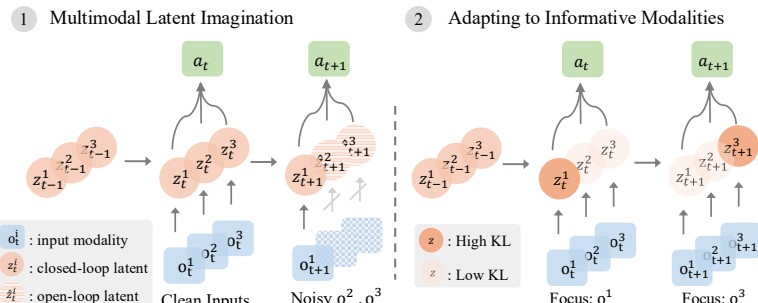

Figure 2: **Model Capabilities.** NOSTRA enables per-modality robustness to noise via latent imagination, and learns latent embeddings that adapt to the heterogeneity in multimodal inputs.

all task phases require every modality, and incorporating irrelevant or unreliable signals can inject noise, corrupt shared representations, and degrade policy performance. Prior work has attempted to address this through task-specific heuristics or modality-specific gating. For example, He et al. (2024) proposed a learned contact predictor to selectively incorporate force/torque feedback only during contact-rich phases, while Du et al. (2022) use information-theoretic criteria to identify useful modalities – but their approach is limited to just two modalities. Despite these advances, existing strategies remain limited in generality and robustness, particularly in real-world settings where multiple modalities may intermittently fail or degrade. This highlights the need for a more scalable framework that can flexibly manage modality reliability while preserving strong visuomotor control.

To address these challenges, we propose a modular latent representation strategy that explicitly partitions the latent space into subspaces dedicated to individual input modalities. This design disentangles modality-specific information, ensuring that the degradation of one modality does not contaminate the entire latent state, while enabling the policy to adaptively weight each input based on its relevance to the task. To remain robust under missing or noisy inputs, we introduce a marginal latent state conditioned on a memory module, which leverages historical context to infer plausible substitutes. Latent variable models have already demonstrated strong capabilities in capturing long-term dependencies for video prediction (Saxena et al., 2021), 3D navigation (Pasukonis et al., 2022), and complex Atari gameplay (Hafner et al., 2022), making them a natural foundation for modeling history in visuomotor control.

In this paper, we present NOSTRA, a multimodal state-space model that enables a visuomotor policy to (1) ignore individual input modalities in case they become noisy, (2) use per-modality latent imagination to do open-loop execution for extended periods of noisy inputs, and (3) selectively attend to informative modalities when learning from datasets that contain non-informative inputs (i.e. heterogeneous). Our visuomotor policy, BC-NOSTRA, trains to fit sequences of expert demonstrations for robotic manipulation tasks. NOSTRA handles each input modality independently, and learns modular representations regularized by an information bottleneck. To allow for our robot policy to maintain task-specific behavior both with and without inputs, we leverage a learned marginal (prior) latent distribution, or *open-loop* latent, for open-loop rollouts, and the variational approximation to the input-conditioned posterior (for each modality) for *closed-loop* rollouts. As we show in our experiments, *multimodal latent imagination* (MLI) allows BC-NOSTRA to be robust against out-of-distribution noises in RGB images (such as occlusion or changing textures in the scene), depth, and robot proprioception. NOSTRA enables efficient pre-training and co-training with large-scale heterogeneous datasets that may contain non-informative inputs. We also deploy BC-NOSTRA on a real robot showing sample-efficient policy learning from just 30 human-teleoperated demonstrations, as well as robustness to camera occlusion. In summary, our contributions are as follows:

1. We present NOSTRA, a novel observation trunk for BC that learns *per-modality* stochastic latents for both *closed-loop* (input-conditioned) and *open-loop* (marginal) states, allowing for action generation with missing sensor inputs using *multimodal latent imagination* (MLI), and efficient learning from heterogeneous datasets that contain non-informative inputs.

2. We create an adaptive mechanism to tackle noisy input modalities, called AdaMLI, that uses the per-modality KL divergence between inferred and imagined latents as a metric to identify noise, and adaptively switches between latent imagination and closed-loop prediction.

3. We train BC-NOSTRA on 12 MuJoCo-based tasks (up to 3 noisy evaluation variants, and 6 modalities including RGB, depth, and robot proprio), showing our method identifies and ignores noisy RGB, depth, and low-dim inputs, achieving higher success by up to 60%, >20% on avg. (Table 1 & 2), in a suite of high-precision, prehensile, and long-horizon tasks.

4. We analyze how BC-NOSTRA trains on tasks where not all inputs are informative (heterogeneous). When pre-training on a dataset with non-informative RGB inputs, BC-NOSTRA is able to better utilize a finite network capacity by re-attributing extracted nats to informative modalities, resulting in better fine-tuning with higher success by up to 18% (Table 4), as well as better co-training with diverse datasets ($\mathcal{O}(10k)$ demos), by up to 20% (Table 5).

## 2 RELATED WORKS

**Behavior Cloning from Diverse Datasets.** Learning visuomotor policies from offline data has become an increasingly popular method of training robots to learn manipulation strategies. These end-to-end policies, trained on expert demonstrations, take multimodal observations as input and output robotic actions (Levine et al., 2016). Lack of standardized large datasets pose significant challenges in learning stable visuomotor policies. Various works have studied leveraging different visual input representations to aid learning, including video (Liang et al., 2024; Hu et al., 2024; Jain et al., 2024; Luo & Du, 2024), voxel-grids (Shridhar et al., 2022; Liu et al., 2024), and point-clouds (Zhu et al., 2024; Peri et al., 2024) to bridge the gap in datasets. Moreover, different generative modeling techniques, such as diffusion models (Chi et al., 2023; Ze et al., 2024; Chen et al., 2024; Pearce et al., 2023; Saxena et al., 2024), flow-matching (Chisari et al., 2024; Zhang & Gienger, 2024; Zhang et al., 2025a; Ding et al., 2024), have been explored. However, significant challenges in robot learning still remain that stem in the inherent heterogeneity of tasks and datasets, which make it hard to learn causal relationships between available inputs and target actions. Current methods are either constrained to specific input modalities (Kim et al., 2024) or simply fuse different modalities (e.g., by feature concatenation), which limit their performance to certain tasks or are sensitive to perception noise. In this work, we propose a general multi-modality visuomotor policy framework that learns a modular per-modality latent space, explicitly regularized by history-conditioned marginal latents, enabling our method to process multimodal inputs robustly and efficiently.

**Autoregressive Models in Robot Learning.** In recent years, autoregressive models for robotic policy learning (Mandlekar et al., 2021; Chen et al., 2021; Gong et al., 2024; Zhang et al., 2025b; Jia et al., 2024) have gained significant attention due to their simple yet scalable architecture designs. Recent works have also identified memory modeling as a significant challenge in state-of-the-art visuomotor policies (Torne et al., 2025) that use fixed observation chunks to condition action generation (Chi et al., 2023). Latent variable models on the other hand have been successful at modelling world dynamics to predict long-horizon video (Saxena et al., 2021; Denton & Fergus, 2018) or for long-horizon planning (Hafner et al., 2019; 2022). In this work, we aim to bring the benefits of state-space models to robot learning, using latent imagination for robust policy execution, while solving practical challenges in robot learning. Specifically, we build upon Hafner et al. (2019) to create a multimodal latent-space with per-modality open-loop prediction capability, that allows our model to be robust to individual inputs. In contrast to (Hafner et al., 2019), our method is trained purely on gradients from action prediction, without reconstructing observations.

## 3 METHOD

We present NOSTRA, a multimodal state-space model that learns a modular latent space given inputs from multiple observation modalities. A modular latent space allows our downstream visuomotor policy, BC-NOSTRA, to attend to useful modalities during rollouts and ignore non-informative ones when training on heterogeneous datasets. BC-NOSTRA can also ignore individual noisy input modalities and use latent imagination to execute with only partial inputs. In this section, we describe our methodology for training BC-NOSTRA to fit a dataset of $N$ expert demonstrations, $\mathfrak{D} = \{^{(i)}o_{1:T}^{1:M}, ^{(i)}a_{1:T}^*\}_{i=1}^N$, containing $M$ different input modalities that include multi-view RGB

Figure 3: **Training BC-NOSTRA.** (Left) We show a graphical model representing the generation ($p_\theta$, solid arrows) and inference ($q_\phi$, broken arrows) procedures in our generative model. (Right) We show details of training the modular latent space in NOSTRA. We use a GRU to aid latent dynamics computation (open-loop latents), with separate networks for processing inputs (closed-loop latents) that share the GRU. Our loss weighs $\mathcal{D}_{\mathrm{KL}}$ and $\mathcal{D}_{\mathrm{NLL}}$, with $\mathcal{D}_{\mathrm{KL}}$ pulling open- and closed-loop gradients together, while $\mathcal{D}_{\mathrm{NLL}}$ ensures acurate action prediction using closed-loop gradients.

images, depth maps, and low-dimensional robot proprioception. We consider each input that is either captured by a separate physical sensor, occupies a different subspace, or requires a separate encoder, as its own modality, since it acts as an independent source of state information.

### 3.1 NOSTRA: A MULTIMODAL STATE-SPACE MODEL

**Joint distribution** BC-NOSTRA learns to maximize the likelihood of data under a joint model $p(a_{1:T}, o_{1:T}^{1:M})$, with learned latents $z_{0:T}^{1:M}$, that is factorized into two components: (1) a latent dynamics model that transitions a collection of latents $z_t^{1:M}$ given the latent history, and (2) an action and observation (not trained) generation model conditioned on the current multimodal latent. Concretely,

$$
\begin{aligned}
p_\theta(a_{1:T}, o_{1:T}^{1:M}) &= \int p_\theta(a_{1:T}, o_{1:T}, z_{0:T}^{1:M}) dz_{0:T}^{1:M} \\
&\triangleq \int p_\theta(z_0^{1:M}) \prod_{t=1}^{T} \underbrace{p_\theta(a_t | z_t^{1:M})}_{\text{action decoder}} p_\theta(o_t | z_t^{1:M}) \underbrace{p_\theta(z_t^{1:M} | z_{t-1}^{1:M})}_{\text{latent dynamics}} dz_{0:T}^{1:M}.
\end{aligned}
\tag{1}
$$

We model all distributions as diagonal multivariate Gaussians with learned means and variances for all latents, and learned means and fixed variance for decoders. Since the focus of this work is to learn visuomotor policies, we set the variance of the observation decoder $p_\theta(o_t | z_t^{1:M})$ to inf, resulting in a model that trains to generate actions only. Note that unlike BC-LSTM (Mandlekar et al., 2021) or Diffusion Policy (Chi et al., 2023), which only learn $p(z_t | o_t)$, we explicitly learn a latent prior $p(z_t^{1:M} | z_{t-1}^{1:M})$ so the policy can operate when some or all observations are missing. Moreover, this joint formulation provides a principled variational inference view to per-modality encodings which define an approximate posterior $q_\phi(z_t^{1:M} | z_{t-1}^{1:M}, o_t)$ as we shall see below.

**Inference** Similar to (Rezende et al., 2014; Kingma & Welling, 2022), we introduce a learned posterior over latents amortized by current observations, $q(z_t^{1:M} | z_{t-1}^{1:M}, o_t^{1:M})$, to help integrate over the space of introduced latents. Since our latent space is modular, we can even sample a subset of latents in this *closed-loop* fashion, say, $z_t^{1:m} \sim q_\phi(z_t^{1:m} | z_{t-1}^{1:M}, o_t^{1:m})$ where $m \leq M$. During training, we default to inferring latents conditioned on all observations (i.e. $m = M$) to subsequently condition the action decoder and latent dynamics.

**Generation** To sample actions given a multimodal latent $z_{t-1}^{1:M}$, $p_\theta(z_t^{1:M} | z_{t-1}^{1:M})$ transitions the multimodal latent dynamics which is decoded into actions using $p(a_t | z_t^{1:M})$. This formulation allows us to sample actions conditioned solely on the latent state in an *open-loop* fashion without having to observe individual inputs. Since our latent space is modular, we can sample open-loop latents for certain modalities while being closed-loop for others. For example, for some $m < M$, we can sample $z_t^{1:m} \sim q_\phi(z_t^{1:m} | z_{t-1}^{1:M}, o_t^{1:m})$ closed-loop while $z_t^{m+1:M} \sim p_\theta(z_t^{m+1:M} | z_{t-1}^{1:M})$ is open-loop. When open-loop latents $z_t^{m+1:M}$ are used to transition latent dynamics, we call this mechanism

"prior-forcing," which enables *latent imagination*. Overall, we train the following model components:

$$\text{closed-loop latents} : z_t^m \sim q_\phi(z_t^m|z_{t-1}^{1:M}, o_t^m) \quad \forall\, m \in [1, M]$$

$$\text{open-loop latents} : \hat{z}_t^m \sim p_\theta(z_t^m|z_{t-1}^{1:M}) \quad \forall\, m \in [1, M]$$

$$\text{action decoder} : a_t \sim p_\theta(a_t|z_t^{1:M})$$

**Training objective** Since computing the log-likelihood of data in the assumed model (1) is intractable, we use ELBO as our training objective, given as

$$
\ln p_\theta(a_{1:T}^*, o_{1:T}^{1:M}) \geq \sum_{t=1}^{T} \Big( \underbrace{\mathbb{E}_{q_\phi(z_t^{1:M}|z_{t-1}^{1:M}, o_t^{1:M})}\big[\ln p_\theta(a_t^*|z_t^{1:M})\big]}_{\mathcal{D}_{\text{LL}}}
$$
$$
- \underbrace{\mathbb{E}_{q_\phi(z_{t-1}^{1:M}|z_{<t-1}^{1:M}, o_{\leq t-1}^{1:M})}\Big[\text{KL}[q_\phi(z_t^{1:M}|z_{t-1}^{1:M}, o_t^{1:M})||p_\theta(z_t^{1:M}|z_{t-1}^{1:M})]\Big]}_{\mathcal{D}_{\text{KL}}} \Big).
\tag{2}
$$

We derive this objective in Appendix B.1, and illustrate training in Figure 3. Note that we skip the observation reconstruction term since we set the decoder variance to inf, thus choosing the minimal loss needed for policy learning and latent imagination. Finally, our training loss is, $\mathcal{L}(\mathfrak{D}; \theta, \phi) := \mathcal{D}_{\text{NLL}} + \beta \mathcal{D}_{\text{KL}}$, where $\mathcal{D}_{\text{NLL}} := -\mathcal{D}_{\text{LL}}$ and $\beta$ controls regularization during training.

**Implementation details** We process image and depth inputs using a ResNet18 (He et al., 2016) convolutional network, and use low-dim inputs as-is, then pass them into separate MLP layers to compute individual closed-loop latents $q_\phi(z_t^m|z_{t-1}^{1:M}, o_t^m)$ for each modality $m$. Each MLP is followed by two small MLPs to output mean and variance. Similar to (Hafner et al., 2019; Saxena et al., 2021), we implement open-loop latent dynamics $p_\theta(z_t^m|z_{t-1}^{1:M})$ using a GRU (Cho et al., 2014). The GRU network is shared between $p_\theta$ and $q_\phi$. Finally, we fuse per-modality latent samples $z_t^{1:M}$ (computed using the re-parameterization trick), using a multi-head attention Modality Transformer followed by another small MLP. To aid training visual representations, we weigh gradients on $p_\theta$ and $q_\phi$ differently by setting $\hat{\mathcal{D}}_{\text{KL}}(q_\phi, p_\theta) := \gamma \mathcal{D}_{\text{KL}}(q_\phi, \text{stop\_grad}(p_\theta)), +(1-\gamma)\mathcal{D}_{\text{KL}}(\text{stop\_grad}(q_\phi), p_\theta)$ to get final loss as $\hat{\mathcal{L}}(\mathfrak{D}; \theta, \phi) := \mathcal{D}_{\text{NLL}} + \beta \hat{\mathcal{D}}_{\text{KL}}$, where $\gamma \in [0, 1]$ (lower values pushing $p_\theta$ toward $q_\phi$ more than vice-versa, relaxing regularization and preventing posterior collapse). We found $\beta$ and $\gamma$ easy to tune, and $\beta = 10^{-4}$ and $\gamma = 0.1$ to work well in our experiments. Background details are in Appendix A.

## 3.2 Latent Imagination & Adapting to Informative Modalities

**Latent imagination for multimodal robustness.** The modular latent space in BC-NOSTRA helps mitigate single-modality corruptions during rollout, e.g., blocked camera or noisy depth. If a modality $m$ is noisy, we sample open-loop $\hat{z}_t^m \sim p_\theta(z_t^m|z_{t-1}^{1:M})$ and closed-loop $z_t^{-m} \sim q_\phi(z_t^{-m}|z_{t-1}^{1:M}, o_t^k)$ where $-m$ is short for $\{1, \ldots, M\} \setminus \{m\}$. After fusing the open- and closed-loop latents in the Modality Transformer, the action decoder samples $a_t \sim p_\theta(a_t|\hat{z}_t^m, z_t^{-m})$, that is the policy output (see Figure 7). This mechanism allows the robot to stay privy of all informative inputs while staying robust against noisy ones. We call this *multimodal latent imagination*, or MLI. MLI emerges from training on entirely noise-free multimodal inputs, i.e. training to decode closed-loop latents only. In Sec. 4.1, we show that MLI strictly strengthens the robustness of the visuomotor policy, compared to baselines that use noisy inputs as-is, or ablations that swap the joint latent state for an imagined one.

**Adaptive multimodal latent imagination** To deploy BC-NOSTRA with multimodal robustness, we require a metric to identify unseen noises in each modality. We use the per-modality KL divergence $\mathcal{D}_{\text{KL}}^m(t) := \text{KL}[q_\phi(z_t^m|z_{t-1}^{1:M}, o_t^m)||p_\theta(z_t^m|z_{t-1}^{1:M})]$ to identify such noises. We build a mechanism based on the intuitive observation that a sharp rise in $\mathcal{D}_{\text{KL}}^m(t)$ suggests modality $m$ is noisy, using which we decide whether to use open- or closed-loop latent for each modality at each timestep. We call this mechanism AdaMLI. Concretely, at timestep $T$, AdaMLI maintains a running average of KL changes $r(T, k_c) = \frac{1}{k_c} \sum_{t=T-k}^{T} |\mathcal{D}_{\text{KL}}^m(t) - \mathcal{D}_{\text{KL}}^m(t-1)|$ up to $k_c$ steps in the past. BC-NOSTRA switches from closed- to open-loop latents for a modality if $\mathcal{D}_{\text{KL}}^m(T) - \mathcal{D}_{\text{KL}}^m(T-1) > f_o * r(T, k)$, since this large change in KL would be attributed to a large unseen noise in that modality. To switch back to closed-loop latents, it resets the running average $r(T, k_o)$, and switches latents if $\mathcal{D}_{\text{KL}}^m(T-1) - \mathcal{D}_{\text{KL}}^m(T) > f_c * r(T, k)$, i.e. if the per-modality KL drops significantly. Note that we compute both latents at each timestep to compute the KL, i.e. we always observe the input, but may or may not route extracted information to the action decoder. Thresholds and running window sizes were easy to tune, and we found $f_o = 20, k_o = 10, f_c = 2, k_c = 2$ to work well for all experiments.

Table 1: **MLI leads to visual robustness.** Showing efficacy of multimodal latent imagination (MLI) on RoboMimic tasks each with 4 different test-time setups: (1) *no noise*, (2) *mask*: square black mask is applied to the camera observations, (3) *cam jitter*: a random perturbation is applied to the camera pose, (4) *table tex*: the texture of the table is changed to an unseen one. We show peak performance in 600 epochs of training, averaged over 50 rollouts. Best performance within $5\%$ is in bold.

| Method | Stack D1 | | | | Stack Three D1 | | | |
|---|---|---|---|---|---|---|---|---|
| | *no noise* | *mask* | *cam jitter* | *table tex* | *no noise* | *mask* | *cam jitter* | *table tex* |
| BC-LSTM (Mandlekar et al., 2021) | **100** | 2 | 10 | 2 | 86 | 20 | 6 | 14 |
| Diffusion Policy (Chi et al., 2023) | **100** | 44 | 72 | 64 | **90** | 66 | 68 | 60 |
| Diffusion Forcing (Chen et al., 2024) | 76 | 50 | 70 | 48 | 22 | 8 | 16 | 2 |
| Diffusion Forcing + prior forcing | | 72 | 72 | 72 | | 18 | 18 | 18 |
| BC-RSSM | **100** | 28 | 42 | 0 | 86 | 38 | 66 | 0 |
| BC-RSSM + LI | | 62 | 62 | 62 | | 64 | 64 | 64 |
| BC-NOSTRA (ours) | **100** | 22 | 24 | 10 | **90** | 68 | 66 | 34 |
| BC-NOSTRA + MLI (ours) | | **84** | **84** | **84** | | **80** | **80** | **80** |

| Method | Square D2 | | | | Coffee D2 | | | |
|---|---|---|---|---|---|---|---|---|
| | *no noise* | *mask* | *cam jitter* | *table tex* | *no noise* | *mask* | *cam jitter* | *table tex* |
| BC-LSTM (Mandlekar et al., 2021) | 46 | 2 | 6 | 2 | 62 | 6 | 20 | 4 |
| Diffusion Policy (Chi et al., 2023) | **58** | 8 | 26 | **40** | 70 | 8 | 16 | 22 |
| Diffusion Forcing (Chen et al., 2024) | 6 | 0 | 2 | 0 | 8 | 0 | 0 | 0 |
| Diffusion Forcing + prior forcing | | 4 | 4 | 4 | | 4 | 4 | 4 |
| BC-RSSM | 48 | 14 | 12 | 20 | 66 | 0 | 20 | 8 |
| BC-RSSM + LI | | 20 | 20 | 20 | | 26 | 26 | 26 |
| BC-NOSTRA (ours) | **54** | 18 | 22 | 0 | **74** | 0 | 16 | 2 |
| BC-NOSTRA + MLI (ours) | | **38** | **38** | **38** | | **38** | **38** | **38** |

**Adapting to informative modalities**  With robotic datasets, not all observations $o^1, ..., o^M$ are equally informative for different tasks or even different phases of completing a task (i.e. heterogeneous). For instance, during the grasping phase of the *coffee* task (see Figure 5), precise manipulation requires the model to focus on inputs from the wrist-view camera rather than the broader agent-view. With a single latent space representing all modalities, their individual influence would need to be learned and expressed in the latent. In contrast, BC-NOSTRA, with modular latents, allows the decoder to focus its learning on extracting the appropriate amount of information from each modality for the action decoding at hand, making learning more sample efficient. Moreover, as we show in Sec. 4.1, this enables our model to learn efficiently from large datasets containing non-informative inputs, in turn enabling better performance when co-training with diverse datasets.

## 4 EXPERIMENTS

We seek to validate four key hypotheses. **H1**: Multimodal Latent Imagination (MLI) in NOS-TRA enables robustness against noisy inputs achieving higher success in manipulation tasks. **H2**: NOSTRA can identify when noises occur, and adaptively employ MLI over RGB, depth, and robot proprioception (AdaMLI). **H3**: Multimodal state-space enables NOSTRA to effectively pretrain from hetereogenous datasets that contain extra modalities uninformative to a downstream task. **H4**: BC-NOSTRA can be deployed on a real robot, with robustness capabilities useful in real-world manipulation. We evaluate on 12 simulated benchmark tasks and deploy NOSTRA to a real robot with 4 challenging manipulation tasks.

**Tasks.**  We evaluate BC-NOSTRA on 12 MuJoCo-based tasks in simulation, including 4 Mimic-Gen (Mandlekar et al., 2023) tasks (*Stack D1*, *Stack Three D1*, *Square D2*, and *Coffee D2*) that contain RGB and robot proprioception, and 8 MimicLabs (Saxena et al., 2025) tasks (*bin carrot, bin bowl, open drawer, close drawer, open microwave, close microwave, open drawer & place bowl, place bowl & close drawer*) that also contain depth maps. We use 1000 demonstrations for each MimicGen task and 200 for MimicLabs tasks, made available by those works. We also show co-training experiments on 2 tasks from the MimicLabs benchmark. Details of all simulated tasks are in Appendix E. Our policies are trained on up to 6 input modalities: agent-view RGB, wrist-view RGB, agent-view depth, end-effector (EEF) position, EEF orientation, and gripper width. On a real Franka Emika Panda robot, we experiment on 4 manipulation tasks: *lift block*, *serve snack*, *marker in cup*, and *pour beans*.

Table 2: **AdaMLI for multimodal noises.** We show success rates (SR) on 8 MimicLabs tasks with unseen noising in different sets of multimodal inputs: "RGB" (masking agent-view and wrist-view images), "RGBD" (additional masking depth maps), and "All" (additionally zero-ing out robot proprioception to emulate sensor failure). BC-NOSTRA measures noise in individual inputs and adaptively employs MLI to maintain high success for this suite of prehensile, non-prehensile, and long-horizon tasks.

| Method | Avg. SR↑ | Avg. Rank↓ | bin carrot | | | bin bowl | | | open drawer | | | close drawer | | |
|---|---|---|---|---|---|---|---|---|---|---|---|---|---|---|
| | | | RGB | RGBD | All | RGB | RGBD | All | RGB | RGBD | All | RGB | RGBD | All |
| BC-LSTM (Mandlekar et al., 2021) | 63.8 | 3.1 | 52 | 30 | 30 | 74 | 76 | 80 | 100 | 100 | 100 | 100 | 100 | 100 |
| Diffusion Policy (Chi et al., 2023) | 61.3 | 3.9 | 40 | 40 | 46 | 56 | 54 | 60 | 92 | 94 | 98 | 98 | 98 | 98 |
| BC-RSSM | 59.9 | 3.4 | 64 | 42 | 52 | 88 | 96 | 90 | 92 | 90 | 92 | 46 | 58 | 60 |
| BC-NOSTRA (ours) | 70.7 | 2.7 | 74 | 76 | 78 | 78 | 86 | 82 | 76 | 68 | 86 | 100 | 100 | 100 |
| BC-NOSTRA + AdaMLI (ours) | **86.5** | **1.4** | 86 | 82 | 86 | 92 | 96 | 90 | 92 | 100 | 96 | 100 | 100 | 100 |

| Method | open microwave | | | close microwave | | | open drawer & put bowl | | | put bowl & close drawer | | |
|---|---|---|---|---|---|---|---|---|---|---|---|---|
| | RGB | RGBD | All | RGB | RGBD | All | RGB | RGBD | All | RGB | RGBD | All |
| BC-LSTM (Mandlekar et al., 2021) | 60 | 52 | 46 | 46 | 46 | 56 | 44 | 56 | 56 | 40 | 40 | 46 |
| Diffusion Policy (Chi et al., 2023) | 44 | 48 | 50 | 68 | 62 | 64 | 22 | 18 | 14 | 68 | 70 | 70 |
| BC-RSSM | 88 | 68 | 82 | 52 | 34 | 20 | 40 | 54 | 40 | 22 | 36 | 32 |
| BC-NOSTRA (ours) | 52 | 70 | 58 | 88 | 80 | 84 | 64 | 60 | 48 | 28 | 26 | 34 |
| BC-NOSTRA + AdaMLI (ours) | 76 | 66 | 78 | 80 | 86 | 88 | 72 | 58 | 78 | 98 | 86 | 90 |

We collect 30 demonstrations for each task using a Meta Quest 2 headset and controller. More details about real-robot tasks are in Appendix H.

**Baselines**   For comparison, we use BC-LSTM (Mandlekar et al., 2021), Diffusion Policy (Chi et al., 2023), Diffusion Forcing (Chen et al., 2024) as baselines in our experiments. We also construct a baseline without modular latents but with the latent imagination capability on the joint latent. Since this baseline shares most architecture design choices with the recurrent state-space model (RSSM) in Hafner et al. (2019), we call it BC-RSSM. More details are in Appendix C.

**Training Details**   We train all models using the Adam (Kingma & Ba, 2014) optimizer with a fixed learning rate of $1e-4$, and Diffusion Policy using the AdamW (Loshchilov & Hutter, 2019) optimizer using the same learning rate with a half-cosine decay schedule (as in the original work). We train all models on a single NVIDIA A40 GPU. Please see Appendix D for all training hyperparameters.

## 4.1   MAIN RESULTS

**Multimodal latent imagination in NOSTRA enables robustness against unseen visual noises.** We evaluate models on 16 distinct task instances created by adding 3 types of visual noise to four simulated tasks: black masks for camera occlusion, random camera position jitter, and changing table textures. Noise is added for multiple durations to hinder performance at critical stages of the task (details in Appendix F). We summarize results in Table 1. On the base tasks with *no noise*, BC-NOSTRA performs comparably to or better than BC-LSTM and Diffusion Policy. When facing noisy variants, we find that latent imagination (LI) significantly boosts policy robustness. Specifically, BC-NOSTRA with multimodal latent imagination (MLI) surpasses Diffusion Policy by up to 30% on *Square D2* and *Coffee D2* and by an impressive 40% on *Stack D1* on masking noise. BC-NOSTRA +MLI outperforms BC-RSSM+LI by up to 22% (∼17% on average), demonstrating the advantage of a separable multimodal latent space that effectively filters out information from noisy inputs.

**Adaptive latent imagination enables robustness against multimodal noise.**   While our initial tests showed strong robustness to purely visual noise using MLI, they relied on an "oracle" to detect when noise occurred - a non-scalable assumption. To address this, we developed AdaMLI, a mechanism that automatically identifies noise across any modality and adaptively switches to open-loop latents to maintain robustness. We tested BC-NOSTRA +AdaMLI on 8 MimicLabs tasks with noise added to RGB images, depth maps, and robot proprioception. Results in Table 2 show that BC-NOSTRA+AdaMLI significantly outperforms all other methods, beating Diffusion Policy by more than 20% on average and up to 60% on long-horizon tasks. This demonstrates its ability to handle noise irrespective of modality, offering a truly scalable solution. We include analysis on the choice of AdaMLI parameters $(k_o, f_o, k_c, f_c)$ in Appendix G.

**BC-NOSTRA outputs task-relevant actions in the absence of image inputs for extended durations.** We analyzed the quality of action trajectories predicted using MLI. Figure 4 shows the $(x, y, z)$

Table 3: **Adapting to informative modalities.** We compare information extracted from two datasets for each task: one with a clear, informative "good view" camera and another with an "occluded view" where the robot blocks the camera, using KL (in nats) after 200 epochs of training. BC-NOSTRA's multimodal latent space adaptively reduces the information extracted from the non-informative, occluded view, while increasing focus on other useful inputs.

| Input type | Stack D1 good view | Stack D1 occl. view | Stack Three D1 good view | Stack Three D1 occl. view | Square D2 good view | Square D2 occl. view | Coffee D2 good view | Coffee D2 occl. view |
|---|---|---|---|---|---|---|---|---|
| Wrist-view RGB | 4.066 | 4.847 (+19.2%) | 4.446 | 5.794 (+30.3%) | 4.374 | 4.909 (+12.2%) | 4.349 | 4.684 (+7.7%) |
| Agent-view RGB | 3.033 | 2.401 (-20.8%) | 3.981 | 2.707 (-32.0%) | 2.876 | 2.580 (-10.3%) | 2.954 | 2.476 (-16.2%) |
| End-effector ori. | 0.740 | 1.039 (+40.4%) | 1.025 | 1.820 (+77.5%) | 0.919 | 1.259 (+36.9%) | 0.452 | 0.643 (+42.3%) |
| End-effector pos. | 0.347 | 0.673 (+94.0%) | 0.401 | 1.083 (+169.9%) | 0.326 | 0.598 (+83.5%) | 0.278 | 0.335 (+20.7%) |
| Gripper width | 0.270 | 0.302 (+11.9%) | 0.324 | 0.462 (+42.7%) | 0.304 | 0.348 (+14.2%) | 0.283 | 0.312 (+10.0%) |
| Total | 8.456 | 9.262 (+9.5%) | 10.177 | 11.865 (+16.6%) | 8.800 | 9.693 (+10.1%) | 8.316 | 8.451 (+1.6%) |

Table 4: Success rates (peak performance in 600 epochs, averaged over 50 rollouts) when fine-tuning behavior cloning policies using pre-trained checkpoints, trained for 200 epochs on a dataset containing a non-informative agent-view image. Results show that a modular latent space almost always leads to more successful fine-tuning.

| Num. demos → | Stack D1 10 | Stack D1 20 | Stack D1 50 | Stack Three D1 10 | Stack Three D1 20 | Stack Three D1 50 | Square D2 10 | Square D2 20 | Square D2 50 | Coffee D2 10 | Coffee D2 20 | Coffee D2 50 |
|---|---|---|---|---|---|---|---|---|---|---|---|---|
| BC-LSTM (Mandlekar et al., 2021) | 28 | 62 | 72 | 2 | 2 | 20 | 2 | 4 | 12 | 6 | 18 | 24 |
| BC-RSSM | 14 | 16 | 6 | 2 | 2 | 4 | 0 | 0 | 0 | 0 | 4 | 8 |
| BC-NOSTRA (ours) | 44 | 74 | 90 | 4 | 14 | 22 | 2 | 12 | 12 | 24 | 34 | 46 |

positions of the robot end-effector during rollout on the *Square* task with *mask* noise for $t \in [75, 110]$. We see that the end-effector trajectory using BC-NOSTRA with latent imagination on the image inputs closely matches that when no noise was added. This shows that the open-loop latent maintains task-relevant information even in the absence of image inputs for extended durations (35 timesteps in this case) leading to successful task execution which would otherwise be hindered by visual noise.

**Multimodal latents in NOSTRA ignore non-informative input modalities.** We train BC-NOSTRA on 2 variants of 4 simulated tasks: one using a standard, informative agent-view camera, and another where the robot self-occludes the agent-view making it non-informative. Results in Table 3 show that BC-NOSTRA effectively identifies when the agent-view image is non-informative. It reduces the information stored in the corresponding latent space by up to 32%, measured by the per-modality KL divergence $\mathcal{D}_{KL}^m$ (in nats), while simultaneously increasing the information extracted from the remaining modalities. This demonstrates BC-NOSTRA's ability to adapt its latent representations to ignore non-informative inputs from a diverse set of modalities.

**BC-NOSTRA enables better pre-training with non-informative inputs.** We pre-trained visuomotor policies for 200 epochs on datasets of 1000 expert demos that included an occluded, non-informative agent-view camera. We then fine-tuned these models on 10, 20, 50 demos containing a useful agent-view image. As shown in Table 4, BC-NOSTRA consistently outperforms BC-RSSM (which lacks modular latents) by up to 30% when fine-tuning on just 10 demos. This strong performance demonstrates that the modular latent structure of BC-NOSTRA learns effective representations even from heterogeneous inputs, enabling superior performance on downstream manipulation tasks.

**BC-NOSTRA learns effectively when co-training with diverse large-scale datasets.** We evaluated our policies on two MimicLabs tasks: *bin bowl* and the longer-horizon *open drawer & put bowl*. For each task, we co-trained the policies with two different retrieved datasets: one large and diverse ($\mathcal{O}(10k)$

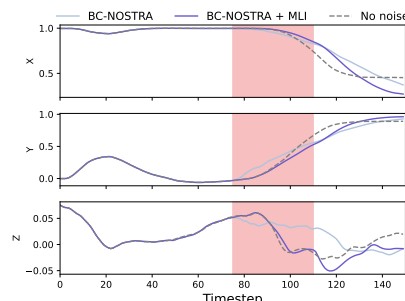

Figure 4: **Multimodal latent imagination (MLI) predicts accurate actions over extended durations.** We noise inputs for $t \in [75, 110]$ (red region) in the *Square* task, and show end-effector trajectories output by BC-NOSTRA with MLI (—) and without (—), as well as BC-NOSTRA on clean inputs (- -). MLI closely matches robot's trajectory to what it could have taken if there was no noise.

Table 5: **Co-training with the MimicLabs Dataset.** We used 10 target demos for the *bin bowl* task and 20 for the *clear table* task. Showing peak performance over 600 epochs of training, averaged over 50 rollouts. *Obj/Skill* refers to retrieving demos with matching objects and skills, +*all* refers to additional retrieval to align camera pose and placement arrangements.

| | bin bowl | | | open drawer & place bowl | | |
|---|---|---|---|---|---|---|
| Method | Target only | Obj/Skill | +all | Target only | Obj/Skill | +all |
| **BC-LSTM** (Mandlekar et al., 2021) | 38 | 52 (+14) | 60 (+22) | 42 | 40 (-2) | 58 (+16) |
| **Diffusion Policy** (Chi et al., 2023) | 10 | 30 (+20) | 30 (+20) | 48 | 28 (-20) | 26 (-22) |
| **BC-RSSM** | 40 | 38 (-2) | 50 (+10) | 40 | 46 (+6) | 38 (-2) |
| **BC-NOSTRA (ours)** | 38 | 44 (+6) | 66 (+28) | 52 | 66 (+14) | 54 (+2) |

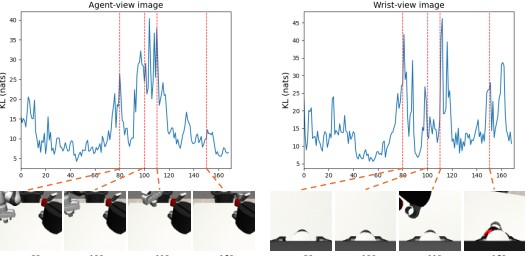

Figure 5: Showing KL divergence (in nats) as a measure of amount of information extracted from each input image for different stages of the *Coffee* task. **Task stages:** $t=80$ - picking up pod; $t=100$ - moving towards coffee machine, agent-view more informative; $t=110$ - coffee machine visible in wrist-view, both views useful; $t=150$ - fine-tuned placing, wrist-view useful but agent-view not.

demonstrations) aligned by object/skill, and a smaller, more specific one ($\mathcal{O}(1k)$ demonstrations) aligned by camera poses and spatial arrangements. Diffusion Policy experienced a significant drop in performance when co-trained with the larger, more diverse dataset on the long-horizon *open drawer & put bowl* task. In contrast, BC-NOSTRA showed a consistent performance boost, achieving a 14% improvement and outperforming a joint latent space model, BC-RSSM, by 8%. Similar gains were observed on the shorter-horizon *bin bowl* task. This consistent improvement highlights the benefits of our modular latent space design in effectively leveraging diverse, heterogeneous data.

## 4.2 REAL-ROBOT EXPERIMENTS

We trained visuomotor policies using BC-NOSTRA, BC-RSSM, and Diffusion Policy on four real-world tasks requiring a variety of motion skills (lifting, picking and placing, pouring), as shown in Figure 9. On three tasks, we show model performance when the camera was occluded by hand, and BC-NOSTRA used AdaMLI to stay robust. Results are in Table 6, averaged over 10 trials.

**BC-NOSTRA learns stable and sample efficient policies in the real world.** Using just 30 human demonstrations, BC-NOSTRA is stable to train and learns policies that outperform Diffusion Policy on 2 out of 4 tasks and is at-par with one other. The modular latent space is crucial to our state-space model as we found BC-RSSM with the joint latent space to undergo unstable training, resulting in no success even on the easy *lift* task. BC-NOSTRA introduces structure to the latent space which acts as regularization resulting in stable downstream performance.

**BC-NOSTRA enables robustness against camera occlusion in the real-world** We evaluated all policies in a real-world setting with an external occlusion of the wrist-view camera by a human hand. As shown in Table 6, BC-NOSTRA with AdaMLI successfully detected and adapted to this unforeseen noise, achieving a success rate that nearly matched its performance without occlusion. In contrast, Diffusion Policy failed to complete two out of three tasks. When the camera was occluded, the robot made jerky motions that caused it to become stuck as errors accumulated. BC-NOSTRA avoided this failure mode by switching to an open-loop latent imagination, effectively ignoring the noisy camera input and maintaining high task success.

Table 6: Success rates (averaged over 10 trials) on four real-robot manipulation tasks.

| Task | lift block | serve snack | | marker in cup | | pour beans | |
|---|---|---|---|---|---|---|---|
| | | no noise | cam occl. | no noise | cam occl. | no noise | cam occl. |
| **Diffusion Policy** (Chi et al., 2023) | 60 | 50 | 0 | 30 | 0 | **70** | 40 |
| **BC-RSSM** | 0 | 20 | 0 | 0 | 0 | 40 | 40 |
| **BC-NOSTRA (ours)** | **60** | **60** | **40** | **50** | **40** | 50 | **50** |

## 5 CONCLUSION

In this paper, we presented BC-NOSTRA, a visuomotor policy that leverages a multi-modal state-space model to enable modality-level robustness to noise, and learns latent embeddings that adapt to the heterogeneity in a manipulation task. We proposed a novel inference strategy called Adaptive Multimodal Latent Imagination (AdaMLI) that allows our visuomotor policy to identify noisy inputs and be robust against them, without any explicit training on action prediction with noisy or missing modalities. Leveraging its modular latent space, BC-NOSTRA offered an effective methodology for fine-tuning visuomotor policies pre-trained or co-trained with diverse robotic datasets. Our experiments strongly indicated that the ideas proposed in this paper can result in robust visuomotor policies, and while proven extensively in single-task settings, our design decisions are general, and we believe will pave the way for the development of robust large-scale multi-task behavior models.

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

# A  BACKGROUND

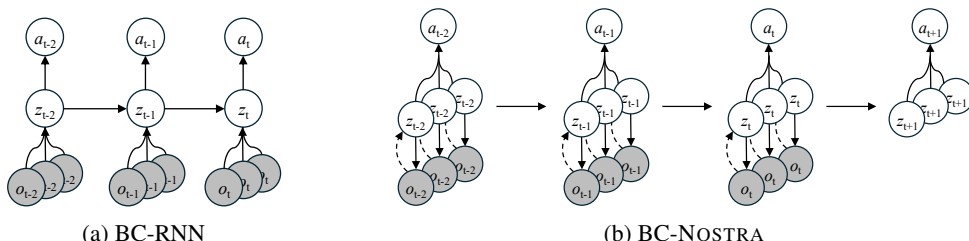

(a) BC-RNN                                   (b) BC-NOSTRA

Figure 6: **Contrasting BC-NOSTRA with BC-RNN (Mandlekar et al., 2021).** Solid arrows represent the generative model, broken and solid arrows combined constitute the inference model.

Given a dataset $\mathfrak{D} = \{^{(i)}o_{1:T},^{(i)} a_{1:T}\}_{i=1}^N$, where $o_{1:T}$ and $a_{1:T}$ denote $N$ paired observation-action trajectories (expert data), $o_t$ is a group of multi-modal observations such as multi-view images and proprioception and $a_t$ is expert action, our goal is to fit a model that can generate actions given observation inputs. Various methods have been proposed in recent literature, that leverage a history of observations to condition action generation at each timestep, and fit this model by maximizing likelihood of $\mathfrak{D}$ in the assumed model. In this section we cover the fundamental approach implemented by (Mandlekar et al., 2021) that uses an RNN to utilize a history of observations (Sec. A.1), and then discuss a latent variable model that uses a combination of stochastic and deterministic latent states for open-loop state-prediction.

## A.1  AUTOREGRESSIVE MODELS FOR BEHAVIOR CLONING

Behavior cloning (BC) is a form of imitation learning where a model learns to replicate expert behavior by mapping observed inputs directly to actions, using supervised learning. BC-RNN (Mandlekar et al., 2021) is an autoregressive imitation learning algorithm that models a visuomotor policy $p(a_{1:T}|o_{1:T})$ factorized as,

$$p(a_{1:T}|o_{1:T}) = \int p(a_{1:T}, z_{0:T}|o_{1:T})dz_{0:T} \triangleq \int p(z_0) \prod_{t=1}^T p(a_t|z_t)p(z_t|z_{t-1}, o_t)dz_{0:T}, \quad (3)$$

where $z_t$ is a latent variable that we introduce, $p(z_t|z_{t-1}, o_t)$ is the observation-conditioned latent dynamics and $p(a_t|z_t)$ is the action decoder. Figure 6a illustrates a typical BC-RNN model. Mandlekar et al. (2021) implemented BC-RNN with a deterministic latent variable $z_t$, using an LSTM (Hochreiter & Schmidhuber, 1997) to compute the latent dynamics, which we call BC-LSTM in this paper. They concatenate learned embeddings of all input modalities into a single latent vector and compute actions using an MLP. In this paper, we build upon the BC-RNN framework for a visuomotor policy where we use a multi-modal state-space model NOSTRA instead of an LSTM.

## A.2  RECURRENT STATE-SPACE MODEL (RSSM)

Recurrent state-space models (RSSMs) are a class of latent-variable world models that factorize the action-conditioned distribution on observations $p(o_{1:T}|a_{1:T})$ as

$$p(o_{1:T}|a_{1:T}) = \int p(o_{1:T}, z_{0:T}|a_{1:T})dz_{0:T} \triangleq \int p(z_0) \prod_{t=1}^T p(o_t|z_t)p(z_t|z_{t-1}, a_t)dz_{0:T}. \quad (4)$$

RSSM uses a combination of stochastic and deterministic variables to represent the latent state $z_t$, with stochastic state filtering information from the current observation and the deterministic state maintaining long-term history using a GRU. In this work, we extend the RSSM architecture to model a multi-modal stochastic state and output actions instead of observations.

## B METHOD DETAILS

### B.1 DERIVATION OF ELBO

BC-NOSTRA models the joint distribution of observations-action trajectories $p(a_{1:T}, o_{1:T}^{1:M})$ by introducing a multimodal latent space $z_t^{1:M}$. We integrate out the introduced latents with the help of a posterior $q(z_t^{1:M}|z_{t-1}^{1:M}, o_t^{1:M})$ which leads us to a lower bound (ELBO) on the true log-likelihood (see Eq. (2)). We prove this variational bound below.

$$\ln p(a_{1:T}, o_{1:T}^{1:M})$$

$$= \ln \int p(a_{1:T}, o_{1:T}, z_{0:T}^{1:M}) dz_{0:T}^{1:M}$$

$$\triangleq \ln \int p(z_0^{1:M}) \prod_{t=1}^{T} p(a_t|z_t^{1:M}) p(o_t|z_t^{1:M}) p(z_t^{1:M}|z_{t-1}^{1:M}) dz_{0:T}^{1:M}$$

$$= \ln \int p(z_0^{1:M}) \prod_{t=1}^{T} p(a_t|z_t^{1:M}) p(o_t|z_t^{1:M}) p(z_t^{1:M}|z_{t-1}^{1:M}) \frac{q(z_t^{1:M}|z_{t-1}^{1:M}, o_t^{1:M})}{q(z_t^{1:M}|z_{t-1}^{1:M}, o_t^{1:M})} dz_{0:T}^{1:M}$$

$$= \ln \mathbb{E}_{q(z_{0:T}^{1:M}|o_{1:T}^{1:M})} \left[ p(z_0^{1:M}) \prod_{t=1}^{T} p(a_t|z_t^{1:M}) p(o_t|z_t^{1:M}) \frac{p(z_t^{1:M}|z_{t-1}^{1:M})}{q(z_t^{1:M}|z_{t-1}^{1:M}, o_t^{1:M})} \right]$$

$$\geq \mathbb{E}_{q(z_{0:T}^{1:M}|o_{1:T}^{1:M})} \left[ \underbrace{\ln p(z_0^{1:M})}_{\text{constant}} + \sum_{t=1}^{T} \ln p(a_t|z_t^{1:M}) + \underbrace{\sum_{t=1}^{T} \ln p(o_t|z_t^{1:M})}_{\text{not trained}} + \sum_{t=1}^{T} \ln \frac{p(z_t^{1:M}|z_{t-1}^{1:M})}{q(z_t^{1:M}|z_{t-1}^{1:M}, o_t^{1:M})} \right]$$

$$= \mathbb{E}_{q(z_{0:T}^{1:M}|o_{1:T}^{1:M})} \sum_{t=1}^{T} \left( \ln p(a_t|z_t^{1:M}) - \frac{q(z_t^{1:M}|z_{t-1}^{1:M}, o_t^{1:M})}{p(z_t^{1:M}|z_{t-1}^{1:M})} \right)$$

$$= \sum_{t=1}^{T} \left( \mathbb{E}_{q(z_t^{1:M}|z_{<t}^{1:M}, o_{\leq t}^{1:M})} \left[ \ln p(a_t|z_t^{1:M}) \right] \right.$$

$$\left. - \mathbb{E}_{q(z_{t-1}^{1:M}|z_{<t-1}^{1:M}, o_{\leq t-1}^{1:M})} \left[ \text{KL}[q(z_t^{1:M}|z_{t-1}^{1:M}, o_t^{1:M})||p(z_t^{1:M}|z_{t-1}^{1:M})] \right] \right)$$

$$\tag{5}$$

We set the initial state $p(z_0^{1:M})$ to be deterministically zeros, and do not train them. Also note that we skip the observation reconstruction term for training, which is equivalent to setting the decoder variance to inf in the ELBO.

## B.2   MULTIMODAL LATENT IMAGINATION

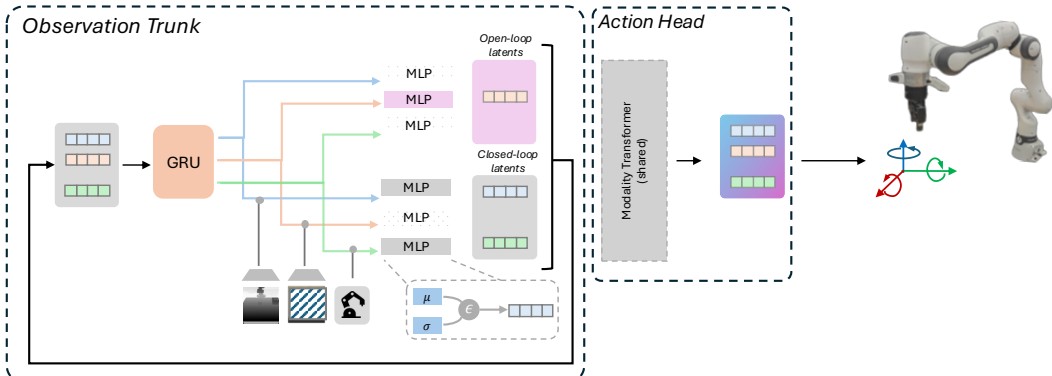

Figure 7: **Multimodal Latent Imagination (MLI).** We show architecture details of the modular latent space in NOSTRA and how we utilize it for multimodal latent imagination. We use a GRU to aid latent dynamics computation (open-loop latents), with separate networks for processing inputs into the closed-loop latents, that share the GRU. To drop an input (shown with slanted dashes), the model swaps its closed-loop latent with the corresponding open-loop latent. A shared Modality Transformer consolidates all latents that are then decoded into actions.

## C BASELINES

### C.1 DIFFUSION FORCING (CHEN ET AL., 2024)

**Original Implementation.** The official implementation of Diffusion Forcing is provided at https://github.com/buoyancy99/diffusion-forcing/tree/paper. In the robot learning setup of this implementation, each diffusion forcing output token consists of two images and 15 actions (a token is the basic generation unit and the diffusion forcing model generates a video by generating a sequence of tokens). The resolution of the synthesized image is set to 32 x 32, hence two images result in a dimension of (32, 32, 6). The official implementation aligns the synthesized action to the dimension of the image such that both images and actions can be together processed by a neural network. Specifically, each action vector will be tiled and repeated to a dimension of (32, 32, 1) in a certain way and then concatenated with the images in the channel dimension. As diffusion forcing leverages action chunking of size 15, the final output dimension is (32, 32, 21).

**Our Implementation in Robosuite.** We implement Diffusion Forcing in our Robosuite tasks based on the original implementation above. To align with common Robosuite setup and enhance computation efficiency, we make two major modifications as described below. First, we set the Diffusion Forcing image size to (84, 84, 3) such that the robot state and target objects are clearly visible. Hence, each output token is in shape (84, 84, 21). Second, as the increased resolution significantly increase the computation requirement, we reduce the depth of the UNet from 4 to 3 and set the batch size to a smaller number (e.g., 16) so that the model can fit onto a single NVIDIA A40 GPU with 48 GB memory.

**Evaluation.** As shown in Table 1, we evaluate Diffusion Forcing in two settings: Diffusion Forcing and Diffusion Forcing *w/ prior-forcing*. In the first setting, following the original implementation, the current observation input is used to update the rolling latent $z$. In the *w/ prior forcing* variant, we only use the current observation to update $z$ if the observation is clean; and will use the predicted rolling latent $\hat{z}$ for action decoding when noisy observation is given. We present examples in Figure 8.

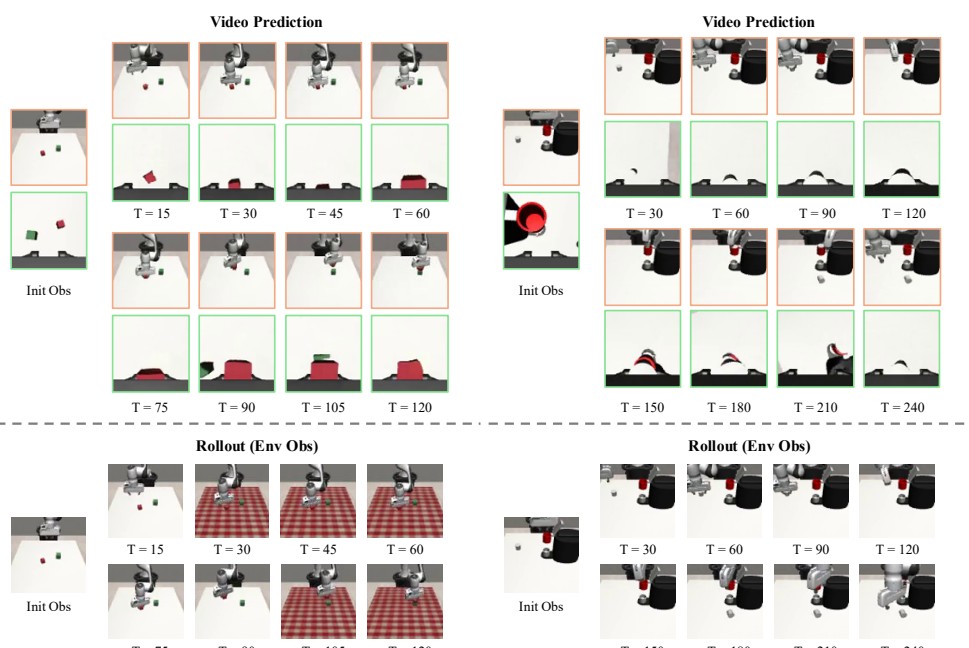

Figure 8: **Evaluating Diffusion Forcing.** Video prediction (top) and rollout observations (bottom) when evaluating the Diffusion Forcing baseline on a *noisy* variant of *Stack D1* (left) *w/ prior forcing*, and the base *Coffee D2* (right) task. When the table texture was changed in *Stack D1*, prior-forcing was able to successfully predict future video and task-relevant actions in an open-loop fashion given the observation history. For the base *Coffee D2* task however video predictions and actions were not accurate enough to complete the task.

## D HYPERPARAMETERS

Table 7: BC-NOSTRA hyperparameters.

| | |
|---|---|
| batch size | 32 |
| sequence length | 16 |
| optimizer | Adam |
| learning rate | $1e-4$ |
| learning rate scheduler | constant |
| num. gradient steps | 300000 |
| image encoder | ResNet18 |
| image resolution | (84, 84) |
| latent size | #modalities * 80 (40 stoch. + 40 det.) |
| modality transformer embed size | 256 |
| modality transformer heads | 2 |
| modality transformer blocks | 1 |
| decoder MLP hidden layers | $400 \times 400$ |
| $\beta$ | $1e-4$ |
| $\gamma$ | 0.1 |
| action space | delta cartesian |

Table 8: BC-LSTM hyperparameters.

| | |
|---|---|
| batch size | 32 |
| sequence length | 16 |
| optimizer | Adam |
| learning rate | $1e-4$ |
| learning rate scheduler | constant |
| num. gradient steps | 300000 |
| image encoder | ResNet18 |
| image resolution | (84, 84) |
| latent size | #modalities * 80 |
| decoder MLP hidden layers | $400 \times 400$ |
| action space | delta cartesian |

Table 9: Diffusion Policy hyperparameters.

| | |
|---|---|
| batch size | 32 |
| observation horizon (To) | 2 |
| action horizon (Ta) | 8 |
| prediction horizon (Tp) | 16 |
| diffusion method | DDPM |
| optimizer | AdamW |
| learning rate | $1e-4$ |
| learning rate scheduler | half-cycle cosine |
| num. gradient steps | 300000 |
| image encoder | ResNet18 |
| image resolution | (84, 84) |
| action space | absolute cartesian |

## E  DETAILS OF SIMULATED ROBOSUITE TASKS

Below are details of MuJoCo-based simulated tasks used in our experiments:

- *Stack D1* (Mandlekar et al., 2023): stack the red block on the green block; blocks are initialized in a 0.4m x 0.4m range with a random yaw rotation
- *StackThree D1* (Mandlekar et al., 2023): stack the red block on the green block, followed by the blue block on the red block; blocks are initialized in a 0.4m x 0.4m range with a random yaw rotation
- *Square D2* (Mandlekar et al., 2023): grasp the nut and insert it onto the peg; nut and peg are initialized in a 0.5m x 0.5m range with random yaw rotations for both nut and peg
- *Coffee D2* (Mandlekar et al., 2023): pick the coffee pod and insert it into the coffee machine; coffee machine is initialized on the right side of the table in a 0.1m x 0.1m range with 90 degrees of yaw rotation variation, pod is initialized on the left side of the table and the pod is initialized in a 0.25m x 0.13m box
- *Bin carrot* (Saxena et al., 2025): put the carrot in the bin; the carrot is initialized in a 0.2m x 0.2m range
- *Bin bowl* (Saxena et al., 2025): put the bowl in the bin; the bowl is initialized in a 0.2m x 0.2m range
- *Open microwave* (Saxena et al., 2025): open the microwave door; microwave is on the left of the robot
- *Close microwave* (Saxena et al., 2025): close the microwave door; microwave is on the right of the robot
- *Open drawer* (Saxena et al., 2025): open the top drawer of the cabinet; cabinet is on the left of the robot
- *Close drawer* (Saxena et al., 2025): close the top drawer of the cabinet; cabinet is on the right of the robot
- *Open drawer & place bowl* (Saxena et al., 2025): open the top drawer of the cabinet and put the bowl in it; the bowl is initialized in a 0.2m x 0.2m range

## F  DETAILS OF NOISY ROBOSUITE TASKS

We provide additional details of noising in different Robosuite (Zhu et al., 2020) tasks from Table 1.

- **Rollout horizon:** We rolled out each task until $t = 600$ timesteps, which corresponds to 30s in real-time using a 20Hz controller.
- **Noising windows:** We choose noising windows roughly corresponding with the critical phases of the task, such as grasping or placing to make the task as difficult as possible during testing. For *Stack D1*, we chose two noising windows, $t \in [25, 75) \cup [100, 150)$, and for *Stack Three D1*, *Square D1*, and *Coffee D2* we chose noising windows $t \in [75, 100) \cup [125, 150)$.

Below are details of different types of noises that we add during noising windows stated above.

- **Masking:** We overlay a black mask of size (50, 50) at a random position on the (84, 84) image, making sure the entire mask was on the image.
- **Camera jitter:** We normally jittered the 3D position of both agent-view and wrist-view cameras centered around their original position with a standard deviation 0.1.
- **Table texture:** We replace the table texture with a new one during the noising windows.

# G    SENSITIVITY ANALYSIS OF ADAMLI PARAMETERS

Below, we show a sensitivity analysis of the AdaMLI parameters ($k_o, f_o, k_c, f_c$) for the *put bowl & close drawer* task from MimicLabs. We choose this task as we showed maximum gains using our per-modality KL-based mechanism for adaptive multimodal latent imagination (AdaMLI) compared to baselines on this task during noisy evaluation in Table 2. The values we used to perform experiments in Table 2 were $k_o = 10, f_o = 20, k_c = 2, f_c = 2$. Below, we show the sensitivity of the model performance (success rates when using BC-NOSTRA + AdaMLI) on each parameter individually, around that choice of parameters.

**Sensitivity of $k_o$ around $k_o = 10, f_o = 20, k_c = 2, f_c = 2$**

| $k_o$ | RGB | RGBD | RGBD+All |
|---|---|---|---|
| 1 | 80 | 80 | 78 |
| 5 | 94 | 80 | **94** |
| **10** | **98** | **86** | 90 |
| 15 | 84 | 80 | 90 |
| 20 | 70 | 56 | 62 |
| Std. Dev. | 11.2 | 11.7 | 13.1 |

**Sensitivity of $f_o$ around $k_o = 10, f_o = 20, k_c = 2, f_c = 2$**

| $f_o$ | RGB | RGBD | RGBD+All |
|---|---|---|---|
| 10 | 86 | **86** | 82 |
| 15 | 84 | 84 | 84 |
| **20** | **98** | **86** | **90** |
| 25 | 80 | 84 | 90 |
| 30 | 90 | 84 | 86 |
| Std. Dev. | 6.8 | 1.1 | 3.6 |

**Sensitivity of $k_c$ around $k_o = 10, f_o = 20, k_c = 2, f_c = 2$**

| $k_c$ | RGB | RGBD | RGBD+All |
|---|---|---|---|
| 1 | 90 | 76 | 80 |
| **2** | **98** | 86 | 90 |
| 4 | 92 | **88** | **92** |
| 6 | 82 | 78 | 74 |
| 8 | 78 | 78 | 82 |
| Std. Dev. | 6.9 | 5.4 | 7.4 |

**Sensitivity of $f_c$ around $k_o = 10, f_o = 20, k_c = 2, f_c = 2$**

| $f_c$ | RGB | RGBD | RGBD+All |
|---|---|---|---|
| 1 | 84 | **88** | 82 |
| **2** | **98** | 86 | **90** |
| 4 | 62 | 58 | 60 |
| 6 | 36 | 42 | 44 |
| 8 | 22 | 42 | 34 |
| Std. Dev. | 31.8 | 22.7 | 24.0 |

The sensitivity analysis, as shown in the tables above for the *put bowl & close drawer* task, confirms our choice of hyperparameters. Our selected configuration of $k_o = 10, f_o = 20, k_c = 2, f_c = 2$ consistently yielded high success rates across the diverse missing-modality scenarios (RGB, RGBD, RGBD+All). Crucially, the standard deviation of success rates remains reasonable for variations around our chosen values of $k_o, f_o,$ and $k_c$, suggesting that the AdaMLI mechanism is relatively robust to small tuning changes for these parameters. However, the standard deviation for the open-to-closed-loop switching parameter, $f_c$, is agreeably high, highlighting its strong influence on the policy's stability and robustness.

## H    REAL-ROBOT TASKS

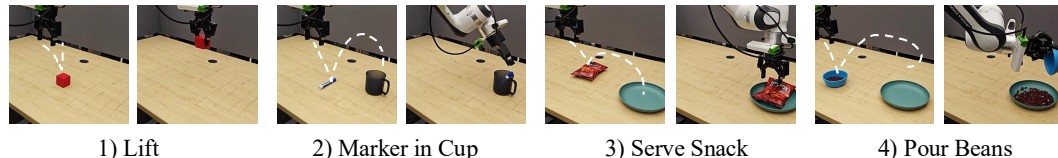

|  1) Lift | 2) Marker in Cup | 3) Serve Snack | 4) Pour Beans |

Figure 9: **Real-world tasks.** We evaluate our method and baselines on 4 tasks using a Franka Emika Panda robot. We show the initial state (left) and final state (right) for each task.

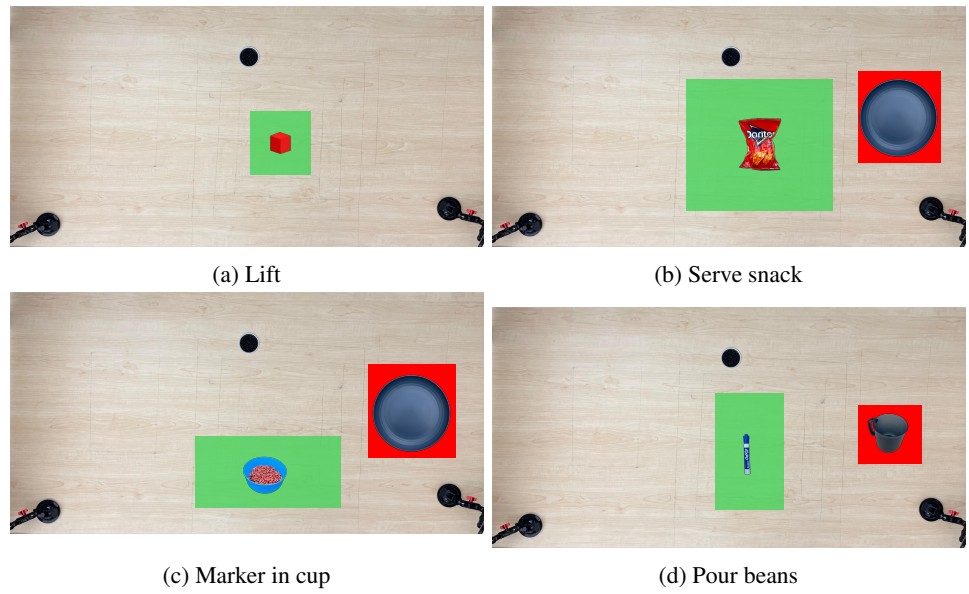

|         (a) Lift         |      (b) Serve snack      |
|        (c) Marker in cup |      (d) Pour beans       |

Figure 10

We run experiments on 4 real-robot tasks:

- *lift*: Lift up the red block. Reset range is 5in × 5in, illustrated in Figure 10a.
- *serve snack*: Pick the snack bag and put it on the plate. The snack is reset in a 10in × 10in range, while the plate had minimal variation. Illustrated in Figure 10b.
- *marker in cup*: Pick up the blue marker and put it in the cup. The marker is reset in a 10in × 5in range and the cup was varied in a smaller 4in x 4in range. Illustrated in Figure 10c.
- *pour beans*: Pour beans from the blue bowl onto the plate. The bowl with beans was reset in a 6in × 10in range while the plate had minimal variation. Illustrated in Figure 10d.

# I LLM USAGE

We used LLMs solely for the purpose of polishing writing. We did not use LLMs for any sort of project ideation or discovering related works.

