# OpenReview forum: "Nostra: Enabling Robust Robot Imitation via Multimodal Latent Imagination"
_ICLR.cc/2026/Conference — Submitted to ICLR 2026_

### Official Review · Reviewer_z4L7 · 2025-10-18

**Soundness:** 3
**Presentation:** 3
**Contribution:** 2
**Rating:** 8
**Confidence:** 4

**Summary:**

This paper introduces NOSTRA, a multimodal state-space model for visuomotor policy learning that enhances robustness in robot imitation learning. Unlike standard BC models that concatenate multimodal features (e.g., RGB, depth, proprioception) into a single latent vector, NOSTRA learns modular per-modality stochastic latent spaces. These latent subspaces allow the policy to adaptively handle missing or noisy inputs by performing multimodal latent imagination.

The authors further propose AdaMLI, a test-time mechanism that uses per-modality KL divergence between inferred and imagined latents to detect noise and selectively switch between open-loop and closed-loop execution.Comprehensive experiments on 12 simulated manipulation tasks (RoboMimic, MimicGen, MimicLabs) and four real-robot tasks demonstrate robustness improvements. The results show that structured, modular latent spaces coupled with adaptive latent imagination enable resilient visuomotor control under multimodal uncertainty.

**Strengths:**

- The proposed method has some level of conceptual innovation. The introduction of per-modality latent subspaces disentangles sensor-specific information, preventing contamination from noisy modalities, which is an improvement over monolithic latent fusion. The proposed latent imagination mechanism (MLI) extends the notion of open-loop predictive world modeling to multimodal BC settings, allowing action inference when certain modalities are unavailable. AdaMLI provides a principled yet practical metric for test-time noise adaptation without retraining or auxiliary supervision.

- Robust empirical validation / Good experimental results: Extensive simulation experiments (Tables 1–5) show consistent and gains over baselines (BC-LSTM, Diffusion Policy, Diffusion Forcing, BC-RSSM). Table 2 and 6 convincingly demonstrate resilience to RGB, depth, and proprioceptive noise as well as real-robot occlusion. Ablations (e.g., with/without MLI, with/without modularity) clearly isolate the contributions of each component.

- Broader and practical impact: Demonstrated robustness in real-robot scenarios with only 30 human demonstrations shows real-world applicability. Training remains data-efficient and uses accessible architectures (ResNet18, GRU).

- The paper is well-written and easy to read. The tables and figures are well-designed and helpful to understand the paper.

**Weaknesses:**

- Incremental and limited novelty in core modeling: While well-executed, the main contribution lies in architectural modularization and the KL-based adaptation heuristic; conceptually, it extends RSSM and variational BC methods rather than introducing a fundamentally new learning paradigm.

- Limited theoretical grounding for AdaMLI: The KL-divergence-based noise metric is intuitive but lacks a formal justification or sensitivity analysis. For example, false positives from high-variance but informative modalities could degrade stability.

- Evaluation scope and generality, minor weakness: Experiments are confined to table-top manipulation; extending to locomotion or force-control tasks would better validate multimodal scalability. All real-robot experiments use a single robot platform (Franka Panda) — it remains unclear if the learned latent structure generalizes across hardware.

- Limited ablation on hyperparameters: The effect of β, γ, and KL thresholds (fo, fc) on policy stability or exploration is not systematically analyzed.

**Questions:**

- Could AdaMLI’s switching mechanism oscillate under mild noise? How stable is it across timesteps and modalities?

- How does the modular latent space scale with the number of modalities (e.g., 10+) — does performance degrade due to GRU bottlenecks?

- Could the per-modality KL divergence be used as an intrinsic exploration signal for self-supervised data collection?

- - Did you try pre-training NOSTRA as a world model (predicting observations) before behavior cloning? Would that improve latent consistency?

---

> ### Author Response · Authors · 2025-11-25
> **Response to Reviewer z4L7 (1/n)**
>
> We thank the reviewer for their time spent in going through our paper, giving insightful comments, and acknowledging that our method is conceptually innovative, simulation experiments are extensive, and our method remains data-efficient in real-robot deployment.
>
> We address their concerns below.
>
> > Incremental and limited novelty in core modeling: While well-executed, the main contribution lies in architectural modularization and the KL-based adaptation heuristic; conceptually, it extends RSSM and variational BC methods rather than introducing a fundamentally new learning paradigm.
>
> Thank you for the thoughtful comments. We agree that our work is related to RSSM and variational BC, but it is more than incremental change because we introduce a new multimodal BC formulation and capability that prior works do not provide. We want to highlight the following points:
>
> 1. Standard RSSM-style world models are built for prediction/planning, and variational BC methods still optimize a conditional policy. In contrast, our goal is to explicitly robustify policy against noisy modalities at test time, or non-informative inputs of the training set. These settings are not addressed by existing BC formulations that assume all inputs are present and useful.
> 2. The key technical step is a modular per-modality latent state-space model that simultaneously learns (i) observation-conditioned posteriors and (ii) history-conditioned priors per modality. This is what makes it possible to “imagine” a missing modality at rollout time, rather than requiring it as input. To our knowledge, no prior BC or RSSM-based policy learns per-modality latent priors explicitly for robust execution.
> 3. AdaMLI is not a bolt-on heuristic, but enabled by the model. The per-modality KL between posterior and prior is only meaningful because the model maintains paired open/closed-loop latents for each modality. Using that KL as a reliability signal yields adaptive routing between sensed and imagined latents online. Prior multimodal BC models cannot do this because they lack modality-wise priors/posteriors.

---

> ### Author Response · Authors · 2025-11-25
> **Response to Reviewer z4L7 (2/n)**
>
> > Limited theoretical grounding for AdaMLI: The KL-divergence-based noise metric is intuitive but lacks a formal justification or sensitivity analysis. For example, false positives from high-variance but informative modalities could degrade stability.
>
> As we stated above, one of the key technical contributions of this work is a modular per-modality latent state-space model that simultaneously learns observation-conditioned posteriors and history-conditioned priors per modality. AdaMLI is enabled by this model formulation, although enabling it does require certain hyperparameters ($k_o, f_o, k_c, f_c$) as we stated in Section 3.2.
>
> We agree with the reviewer that a comprehensive analysis of the AdaMLI mechanism's stability is crucial. Below, we show a sensitivity analysis of the hyperparameters ($k_o, f_o, k_c, f_c$) on the *put bowl & close drawer* task from MimicLabs. We choose this task as we showed maximum gains in noisy evaluation using AdaMLI compared to baselines in this task. The values we used to perform experiments in Table 2 were $k_o = 10, f_o=20, k_c=2, f_c=2$. Below, we show the sensitivity of the model performance (success rates when using BC-NOSTRA + AdaMLI) for each parameter individually, around that choice of parameters.
>
>
> **Sensitivity of $k_o$ around $k_o = 10, f_o=20, k_c=2, f_c=2$**
>
> | $k_o$ | **RGB** | **RGBD** | **RGBD+All** |
> | :---: | :---: | :---: | :---: |
> | 1 | 80 | 80 | 78 |
> | 5 | 94 | 80 | **94** |
> | **10** | **98** | **86** | 90 |
> | 15 | 84 | 80 | 90 |
> | 20 | 70 | 56 | 62 |
> | **Std. Dev.** | **11.2** | **11.7** | **13.1** |
>
> **Sensitivity of $f_o$ around $k_o = 10, f_o=20, k_c=2, f_c=2$**
>
> | $f_o$ | **RGB** | **RGBD** | **RGBD+All** |
> | :--------: | :---------: | :----------: | :--------------: |
> | 10 | 86 | **86** | 82 |
> | 15 | 84 | 84 | 84 |
> | **20** | **98** | **86** | **90** |
> | 25 | 80 | 84 | 90 |
> |30 | 90 | 84 | 86 |
> | **Std. Dev.** | **6.8** | **1.1** | **3.6** |
>
> **Sensitivity of $k_c$ around $k_o = 10, f_o=20, k_c=2, f_c=2$**
>
> | $k_c$ | **RGB** | **RGBD** | **RGBD+All** |
> | :---: | :-----: | :------: | :----------: |
> | 1 | 90 | 76 | 80 |
> | **2** | **98** | 86 | 90 |
> | 4 | 92 | **88** | **92** |
> | 6 | 82 | 78 | 74 |
> | 8 | 78 | 78 | 82 |
> | **Std. Dev.** | **6.9** | **5.4** | **7.4** |
>
> **Sensitivity of $f_c$ around $k_o = 10, f_o=20, k_c=2, f_c=2$**
>
> | $f_c$ | **RGB** | **RGBD** | **RGBD+All** |
> | :---: | :-----: | :------: | :----------: |
> | 1 | 84 | **88** | 82 |
> | **2** | **98** | 86 | **90** |
> | 4 | 62 | 58 | 60 |
> | 6 | 36 | 42 | 44 |
> | 8 | 22 | 42 | 34 |
> | **Std. Dev.** | **31.8** | **22.7** | **24.0** |
>
> The sensitivity analysis, as shown in the tables above for the *put bowl & close drawer* task, confirms our choice of hyperparameters. Our selected configuration of $k_o = 10, f_o=20, k_c=2, f_c=2$ consistently yielded high success rates across the diverse missing-modality scenarios (RGB, RGBD, RGBD+All). Crucially, the standard deviation of success rates remains reasonable for variations around our chosen values of $k_o, f_o, \text{and } k_c$, suggesting that the AdaMLI mechanism is relatively robust to small tuning changes for these parameters. However, the standard deviation for the open-to-closed-loop switching parameter, $f_c$, is agreeably high, highlighting its strong influence on the policy's stability and robustness. We have included these detailed sensitivity results in Appendix G of the revised paper to provide the necessary empirical grounding and to serve as a valuable reference for future work adapting this methodology.

---

> ### Author Response · Authors · 2025-11-25
> **Response to Reviewer z4L7 (3/n)**
>
> > Evaluation scope and generality, minor weakness: Experiments are confined to table-top manipulation; extending to locomotion or force-control tasks would better validate multimodal scalability. All real-robot experiments use a single robot platform (Franka Panda) — it remains unclear if the learned latent structure generalizes across hardware.
>
> Thank you for this valuable comment regarding the scope and generality of our evaluation. We agree that broadening the evaluation to tasks like locomotion or force-control would be excellent for validating multimodal scalability. However, we chose to focus on table-top manipulation because, in the current state of robotics research, it remains one of the most significant and complex challenges, especially concerning reliable visuomotor control under uncertainty. Our method, NOSTRA, is designed to specifically solve challenges within this domain, such as handling sensor dropout and noise in multi-camera setups. We suggest that other areas, such as mobile manipulation, may introduce new challenges (e.g., navigation and state estimation errors) that could conflate the relative advantages and disadvantages of our method and the baselines we compare against, making it harder to isolate the benefits of our method. Finally, our real-robot experiments were limited to the Franka Panda platform due to practical constraints and accessibility to hardware, which is a common limitation in experimental robotics research. We believe the observed robustness in our real-world tests still offers a compelling proof of concept for the real-world applicability and effectiveness of NOSTRA.
>
>
> > Limited ablation on hyperparameters: The effect of β, γ, and KL thresholds (fo, fc) on policy stability or exploration is not systematically analyzed.
>
> We discussed the sensitivity of model performance on AdaMLI parameters ($k_o, f_o, k_c, f_c$) above. Here, we add more discussion around the effects of training parameters, $\beta$ and $\gamma$. As stated in Section 3.1, $\beta$ controls the effect of KL regularization during training. Higher $\beta$ means more weight on training the alignment of prior ($p_{\theta}$, open-loop) and posterior ($q_{\phi}$, closed-loop) latent representations. On the other hand, $\gamma$ controls gradient flow from the KL term in the ELBO into $p_{\theta}$ and $q_{\phi}$.
>
> To understand the effects of $\beta$, we analyzed the action decoding loss and the success rate of the policy during rollouts. Below are these values (after 50 epochs of training) against four different values of $\beta$ we tried, on the *coffee D2* task. We kept $\gamma = 0.5$ (equal gradient flow into both terms of the KL) for this analysis.
>
> $\beta$ | Action NLL $\downarrow$ | Success Rate $\uparrow$ |
> | :---: | :---: | :---: |
> 1 | 0.0133 | 0% |
> 1e-2 | 0.2361 | 0% |
> 1e-3 | 0.0079 | 42% |
> 1e-4 | 0.0079 | 40% |
> 1e-5 | 0.0079 | 20% |
>
> We observe that the action reconstruction loss stagnates after $\beta = 1e-3$ and below, however, lowering $\beta$ beyond $1e-4$ led to significant drop in model performance during rollouts (showing that a certain amount of regularization helps downstream success). This bolsters our choice of $\beta=1e-4$ in our experiments.
>
> $\gamma$ gives us a lever to further tune gradient flow so that we can enhance the quality of action prediction conditioned on open-loop latents ($p_{\theta}$). Hence, we try three values of $\gamma$ ($0.5, 0.3, 0.1$) where values lower than $0.5$ favor training $p_{\theta}$ by minimizing changes in the closed-loop latents ($q_{\phi}$) while boosting gradients for $p_{\theta}$. We analyze the action-reconstruction error when computing actions conditioned on the open-loop latents. (Note that we did not use this error for training.)
>
> $\gamma$ | Action NLL $\downarrow$ | Action recon. error using open-loop latents $\downarrow$ |
> | :---: | :---: | :---: |
> 0.5 | 0.0079 | 2.119 |
> 0.3 | 0.0080 | 2.064 |
> 0.1 | 0.0077 | 1.856 |
>
> While action NLLs are similar, we see that $\gamma = 0.1$ obtains the best action-reconstruction error when computing actions using the learned prior (open-loop latents). This bolsters our claim that an uneven gradient flow into $p_{\theta}$ and $q_{\phi}$ helps obtain better open-loop latent representations which strengthen our multimodal latent imagination (MLI) mechanism for robust execution.
>
> Hence, the ablation studies for $\beta$ and $\gamma$ provide strong empirical justification for our chosen hyperparameter settings.

---

> ### Author Response · Authors · 2025-11-25
> **Response to Reviewer z4L7 (4/n)**
>
> > Could AdaMLI’s switching mechanism oscillate under mild noise? How stable is it across timesteps and modalities?
>
> Thank you for the question. To show how the per-modality KL divergence fluctuates with and without noise, we have shared a video on [**this link**](https://drive.google.com/file/d/1ojZNHrese-7wS30r_lkyQntpBvpxhPuD/view?usp=sharing). In these example evaluations of the *square* task, we add noise to the camera RGB images - the video on the left shows policy execution when just the agent-view camera was jittered, while the video on the right shows that when both agent-view and wrist-view cameras were jittered. Below the image captures, we show the per-modality KL divergence between open- and closed-loop latents for each modality (those for cameras are labeled *agentview* and *robot0_eye_in_hand* respectively). We use these KL divergences in our switching mechanism in AdaMLI.
>
> We make two key observations in this example: (i) while the KL divergences do oscillate during normal execution, in the presence of unseen noises, these fluctuations are significantly larger, and (ii) modalities not affected by noise do not show major fluctuations in KL. The hyperparameters in AdaMLI are set such that oscillations due to changing usefulness of each modality are filtered out, while the fluctuations due to unseen noises are captured. Moreover, only the noise-affected modalities are captured, while other other modalities stay intact for policy execution.
>
> In summary, the design successfully achieves its goal: the AdaMLI mechanism is stable, with hyperparameters set to filter normal operational noise and selectively detect the significant KL divergence shifts caused by genuine unseen input corruption, ensuring robust, adaptive policy execution.
>
>
>
> > How does the modular latent space scale with the number of modalities (e.g., 10+) — does performance degrade due to GRU bottlenecks?
>
> Thank you for bringing up this insightful discussion. The modular latent space design has a direct relationship with the number of modalities. We include a table of hyperparameters in Appendix D which explicitly states the size of the latent space as the number of modalities grows. For all our experiments, this was set to 80 $\times$ number of modalities. That is, the latent size grows linearly with the number of modalities. It is also important to note that for all capacity-controlled experiments, the total latent size of baseline methods with a single joint latent (e.g., BC-LSTM and BC-RSSM) was explicitly matched to be the same as that of our method.
>
> Regarding the potential for GRU bottlenecks: the GRU helps to consolidate the per-modality latents and generate temporal context. However, how this scales with an increasing number of modalities, or the integration of the design decisions in this paper with other recurrent architectures, remains an open question, and we leave that to future work.
>
>
> > Could the per-modality KL divergence be used as an intrinsic exploration signal for self-supervised data collection?
>
> Thank you for the insightful connection. The computed per-modality KL divergence quantifies the epistemic uncertainty about the latent state for that specific modality. Specifically, it represents the difference between the model's prediction (prior/imagined latent) and the observation-conditioned truth (posterior/inferred latent). We believe this could signal when the model is less confident about a particular sensor's input. However, turning this into a reliable intrinsic exploration reward would likely introduce additional challenges, since KL can spike for reasons unrelated to “useful novelty” (e.g., transient sensor noise, occlusions, or distribution shift that the agent cannot resolve), and naive maximization could incentivize visiting states that are simply hard to sense rather than informative. We view this as a promising direction, but it would likely need stabilizing mechanisms (e.g., temporal smoothing, controllability/learnability filters, or coupling with task progress) to avoid these failure modes.

---

> ### Author Response · Authors · 2025-11-25
> **Response to Reviewer z4L7 (5/n)**
>
> > Did you try pre-training NOSTRA as a world model (predicting observations) before behavior cloning? Would that improve latent consistency?
>
>
> Thank you for this highly relevant question. We did explore the effect of pre-training or jointly training NOSTRA as a world model that reconstructs observations. Theoretically, this is a very compelling idea, as generating observations with the posterior latent state would enforce a tighter coupling via KL and improve latent consistency between the prior (imagined) and posterior (inferred) states. This is because the observation reconstruction loss would provide an additional, strong gradient signal to align the open-loop prediction with the closed-loop inference. However, in practice, we found that introducing the observation prediction term alongside the behavior cloning and KL-regularization losses was empirically hard to tune. The large magnitude and complex dynamics of the reconstruction loss often dominated the objective, leading to a degradation in the quality of the action prediction and, consequently, a decrease in the downstream success rates. Given that our primary objective is robust behavior cloning, we opted to remove this auxiliary loss term. This decision is also theoretically justified, as we discuss in the paper, by setting the observation decoder's standard deviation to infinity, which effectively removes the term from the ELBO objective while retaining the structural benefits of the generative world-modeling formulation.
>
>
> *We hope that our response addressed your concerns. If so, we would be very grateful if you would consider adjusting your score to reflect the clarifications and additional analysis provided.*

---

### Official Review · Reviewer_62T8 · 2025-10-27

**Soundness:** 1
**Presentation:** 2
**Contribution:** 3
**Rating:** 4
**Confidence:** 3

**Summary:**

NOSTRA is a novel behaviour cloning approach that makes robot policies more reliable when using multiple sensors, like different cameras, depth, and proprioception. Instead of mixing all sensor information together, it learns a separate latent state for each sensor, so when one sensor becomes noisy or blocked, the model can simply ignore that sensor and use its internal memory to continue acting smoothly. This “latent imagination” idea comes from Dreamer, but here it is applied per-sensor. A simple KL-based rule is used to automatically detect when a sensor becomes unreliable and switch it off. In experiments, this works well—the policy stays stable under occlusions and noise, and also learns better from datasets where some inputs are unhelpful. However, the math in the paper does not fully match the implementation.

**Strengths:**

The main strength of this paper is its clear and effective design choice of separating the latent representation by modality, which allows the policy to selectively ignore unreliable sensors and remain stable even when some inputs are noisy, occluded, or uninformative. This per-modality structure makes the overall approach intuitive and easy to reason about, and the paper demonstrates that it leads to robust performance improvements in practice. The experiments are solid and comprehensive, covering multiple simulated benchmarks, heterogeneous datasets, and real robot evaluations, all showing consistent gains. Additionally, the paper is well written, clearly structured, and easy to follow, making the core ideas and implementation details accessible.

**Weaknesses:**

A limitation of the paper is that the mathematical formulation does not fully match the implementation. The method is framed as a joint generative model with an observation likelihood, but this likelihood is effectively removed by assuming infinite variance, which weakens the theoretical justification of the stated ELBO objective. As a result, the actual training loss functions more as a behavior cloning objective with a KL regularizer, rather than a principled variational bound. In addition, the KL-based modality switching mechanism relies on hand-tuned threshold values, and the paper does not analyze how sensitive the performance is to these choices. Finally, the paper does not provide capacity-controlled ablations to separate the benefits of modular latent factorization from the possibility that the model simply has greater latent capacity; controlling for total parameter count or latent dimensionality would be necessary to isolate the architectural contribution.

**Questions:**

1. Mismatch between model and objective.
   The method is presented as a joint generative model that includes an observation
   likelihood term p(o_t | z_t), but in practice this likelihood is removed by assuming
   infinite variance. This makes it unclear what objective is actually being optimized,
   and whether the final loss should still be interpreted as an ELBO.

2. Why present a joint model if observations are never reconstructed?
   Since p(o_t | z_t) does not contribute to the training loss, the justification for
   including it in the generative model is unclear. It may be more precise to present
   the method directly as a conditional policy p(a_{1:T} | o_{1:T}) with a latent
   bottleneck, instead of a joint model that is not trained as such.

3. Initialization of the latent prior is unspecified.
   Appendix B introduces a prior p(z_0), but the training objective does not include
   a KL(q(z_0) || p(z_0)) term. The paper should clarify how z_0 is initialized and
   whether any prior over z_0 is actually used in practice during training or inference.

4. The KL regularizer is not the ELBO KL.
   The loss uses a stop-gradient KL mixture:
       γ KL(q || stop(p)) + (1 - γ) KL(stop(q) || p)
   This breaks the variational interpretation of the ELBO. The paper should explain
   the motivation behind this surrogate objective and clarify that the optimized loss
   is no longer a strict ELBO.

5. Heuristic modality switching without sensitivity analysis.
   The KL-based modality switching relies on hand-tuned thresholds and hyperparameters.
   The paper should provide sensitivity analysis to demonstrate robustness to these
   choices and to validate the stability of the switching mechanism.

6. Uncontrolled model capacity differences.
   Because each modality has its own latent vector, the total latent capacity may be
   larger than in single-latent baselines. The paper should include capacity-controlled
   comparisons (e.g., matching total latent dimensionality or parameter count) to ensure
   that improvements are due to the modular design rather than simply increased capacity.

Overall:
   The paper would score significantly better if these theoretical issues were clarified
   and linked more directly to the actual training objective and implementation.

---

> ### Author Response · Authors · 2025-11-25
> **Response to Reviewer 62T8 (1/n)**
>
> We thank the reviewer for their time spent in going through our paper, giving insightful comments, and acknowledging that our choice of separating the latent representation by modality is clear and effective, and that our experiments are solid and comprehensive.
>
> We address their concerns below.
>
> > Why present a joint model if observations are never reconstructed? Since p(o_t | z_t) does not contribute to the training loss, the justification for including it in the generative model is unclear. It may be more precise to present the method directly as a conditional policy p(a_{1:T} | o_{1:T}) with a latent bottleneck, instead of a joint model that is not trained as such.
>
> We agree with the reviewer that our training objective does not include an observation reconstruction term. We nevertheless present NOSTRA as a **joint latent-variable model over actions and observations** because this factorization is what enables the capability we target: **a policy that can act either closed-loop (conditioning on any subset of modalities) or open-loop (with no observations)**. Concretely, we assume the standard state-space generative structure
> $$p_\theta(a_{1:T}, o_{1:T}, z_{0:T}) = p(z_0) \prod_t p_\theta(z_t\mid z_{t-1})\ p_\theta(a_t\mid z_t)\ p_\theta(o_t\mid z_t),$$
>
> and then **train only the action-prediction and latent-dynamics parts**, while treating $p_{\theta}(o_t | z_t)$ as an untrained likelihood with fixed variance. In our implementation we set that variance to $\inf$, which makes the observation log-likelihood a constant and removes it from the ELBO, yielding exactly the objective in Eq. (2): action NLL plus a KL term that regularizes inferred (closed-loop) latents toward the history-conditioned prior (open-loop) latents. Formally, our method is equivalent to maximizing the ELBO of a joint model with an observation decoder whose variance is fixed to $\inf$; the decoder is included to define the posterior family but contributes zero gradient.
>
> This joint formulation is not cosmetic. It provides (i) a principled variational inference view, where the per-modality encoder defines an approximate posterior $q_\phi(z_t | z_{t-1}, o_t)$, and (ii) a learned prior $p_\theta(z_t | z_{t-1})$ that can be rolled forward **without observations**. The KL term between these two distributions is exactly what makes “latent imagination” possible and what we later use as a modality-wise reliability signal in AdaMLI. Unlike BC-RNN or Diffusion Policy, which only learn $p(z_t | o_t)$, we explicitly learn a latent prior $p(z_t | z_{t-1})$ so the policy can operate when some or all observations are missing.
>
> Presenting the method solely as a conditional policy $p(a_{1:T} | o_{1:T})$ would obscure this open-loop generative path and the role of the KL alignment in making open-loop and closed-loop behavior consistent. We are happy to add a remark noting this in Section 3.1, but we prefer the joint graphical model because it makes the open-loop prior path and KL-based reliability test explicit.
>
> Finally, we did experiment with training an auxiliary observation reconstruction term, but found no empirical gains, so we excluded it for simplicity. In other words, we keep the joint model as the correct probabilistic scaffold for our latent-imagination policy, while choosing the minimal loss needed for action learning and robustness.
>
>
> > Mismatch between model and objective. The method is presented as a joint generative model that includes an observation likelihood term p(o_t | z_t), but in practice this likelihood is removed by assuming infinite variance. This makes it unclear what objective is actually being optimized, and whether the final loss should still be interpreted as an ELBO.
>
> We appreciate the concern. The objective we optimize is still an ELBO, but for a slightly specialized joint model in which the observation likelihood is explicitly made uninformative. Concretely, our generative scaffold includes $p(o|z)$ but we set its decoder variance to $\inf$. So the observation term contributes no gradient and can be dropped without changing the optimizer. This is exactly what we state in the paper: “we set the variance of the observation decoder … to $\inf$, resulting in a model that trains to generate actions only.” Hence, there is no mismatch in the optimization. To summarize, we have:
>
> 1. Model: joint latent state-space factorization is retained to define a closed-loop posterior and an open-loop prior
> 2. Objective: ELBO of that model under an uninformative observation decoder, yielding action NLL + KL alignment, which is the minimal loss needed for action learning and latent imagination
>
> We make this clearer in the paper by adding a short remark in Section 3.1 of the revised paper, where we define the training objective.

---

> ### Author Response · Authors · 2025-11-25
> **Response to Reviewer 62T8 (2/n)**
>
> > Initialization of the latent prior is unspecified. Appendix B introduces a prior p(z_0), but the training objective does not include a KL(q(z_0) || p(z_0)) term. The paper should clarify how z_0 is initialized and whether any prior over z_0 is actually used in practice during training or inference.
>
> Thank you for pointing this out - we indeed missed out specifying the prior distribution over $z_0$, which is not trained and is fixed to be constant. We set $z_0$ to be deterministically zeros for each modality-wise latent variable. We have updated Appendix B.1 with this information.
>
> > The KL regularizer is not the ELBO KL. The loss uses a stop-gradient KL mixture: γ KL(q || stop(p)) + (1 - γ) KL(stop(q) || p) This breaks the variational interpretation of the ELBO. The paper should explain the motivation behind this surrogate objective and clarify that the optimized loss is no longer a strict ELBO.
>
> Thank you for bringing this up for discussion. We want to emphasize that our mixture KL does not change the value of the KL term, because $\gamma*KL(q_{\phi} || stopgradient(p_{\theta})) + (1 - \gamma) KL(stopgradient(q_{\phi}) || p_{\theta})$ is equal to $KL(q_{\phi} || p_{\theta})$. Wrapping a stop_gradient around $q$ or $p$ does not change the value of the KL. Therefore, the ELBO loss indeed stays intact. What does change though, is how the optimizer computes gradients to minimize this loss, which has been used in previous works such as [1] and [2] for balancing gradient flow between learned prior and posterior distributions.
>
> $\gamma$ gives us a lever to tune gradient flow so that we can enhance the quality of action prediction conditioned on open-loop latents ($p_{\theta}$). To analyze its effects, we trained policies on the *coffee D2* task with three values of $\gamma$ ($0.5, 0.3, 0.1$), where values lower than $0.5$ favor training $p_{\theta}$ by minimizing changes in the closed-loop latents ($q_{\phi}$) while boosting gradients for $p_{\theta}$. We analyze the action-reconstruction error when computing actions conditioned on the open-loop latents. (Note that we did not use this error for training.)
>
> $\gamma$ | Action NLL $\downarrow$ | Action recon. error using open-loop latents $\downarrow$ |
> | :---: | :---: | :---: |
> 0.5 | 0.0079 | 2.119 |
> 0.3 | 0.0080 | 2.064 |
> 0.1 | 0.0077 | 1.856 |
>
> While action NLLs are similar, we see that $\gamma = 0.1$ obtains the best action-reconstruction error when computing actions using the learned prior (open-loop latents). This bolsters our claim that an uneven gradient flow into $p_{\theta}$ and $q_{\phi}$ helps obtain better open-loop latent representations which strengthen our multimodal latent imagination (MLI) mechanism for robust execution.

---

> ### Author Response · Authors · 2025-11-25
> **Response to Reviewer 62T8 (3/n)**
>
> > Heuristic modality switching without sensitivity analysis. The KL-based modality switching relies on hand-tuned thresholds and hyperparameters. The paper should provide sensitivity analysis to demonstrate robustness to these choices and to validate the stability of the switching mechanism.
>
> Thank you for the thoughtful remark. We agree that a comprehensive analysis of the Adaptive Multimodal Latent Imagination (AdaMLI) mechanism's stability is crucial. Below, we show a sensitivity analysis of the hyperparameters ($k_o, f_o, k_c, f_c$) on the *put bowl & close drawer* task from MimicLabs. We choose this task as we showed maximum gains using our per-modality KL-based mechanism for adaptive multimodal latent imagination (AdaMLI) compared to baselines on this task during noisy evaluation. The values we used to perform experiments in Table 2 were $k_o = 10, f_o=20, k_c=2, f_c=2$. Below, we show the sensitivity of the model performance (success rates when using BC-NOSTRA + AdaMLI) for each parameter individually, around that choice of parameters.
>
> **Sensitivity of $k_o$ around $k_o = 10, f_o=20, k_c=2, f_c=2$**
>
> | $k_o$ | **RGB** | **RGBD** | **RGBD+All** |
> | :---: | :---: | :---: | :---: |
> | 1 | 80 | 80 | 78 |
> | 5 | 94 | 80 | **94** |
> | **10** | **98** | **86** | 90 |
> | 15 | 84 | 80 | 90 |
> | 20 | 70 | 56 | 62 |
> | **Std. Dev.** | **11.2** | **11.7** | **13.1** |
>
> **Sensitivity of $f_o$ around $k_o = 10, f_o=20, k_c=2, f_c=2$**
>
> | $f_o$ | **RGB** | **RGBD** | **RGBD+All** |
> | :--------: | :---------: | :----------: | :--------------: |
> | 10 | 86 | **86** | 82 |
> | 15 | 84 | 84 | 84 |
> | **20** | **98** | **86** | **90** |
> | 25 | 80 | 84 | 90 |
> |30 | 90 | 84 | 86 |
> | **Std. Dev.** | **6.8** | **1.1** | **3.6** |
>
> **Sensitivity of $k_c$ around $k_o = 10, f_o=20, k_c=2, f_c=2$**
>
> | $k_c$ | **RGB** | **RGBD** | **RGBD+All** |
> | :---: | :-----: | :------: | :----------: |
> | 1 | 90 | 76 | 80 |
> | **2** | **98** | 86 | 90 |
> | 4 | 92 | **88** | **92** |
> | 6 | 82 | 78 | 74 |
> | 8 | 78 | 78 | 82 |
> | **Std. Dev.** | **6.9** | **5.4** | **7.4** |
>
> **Sensitivity of $f_c$ around $k_o = 10, f_o=20, k_c=2, f_c=2$**
>
> | $f_c$ | **RGB** | **RGBD** | **RGBD+All** |
> | :---: | :-----: | :------: | :----------: |
> | 1 | 84 | **88** | 82 |
> | **2** | **98** | 86 | **90** |
> | 4 | 62 | 58 | 60 |
> | 6 | 36 | 42 | 44 |
> | 8 | 22 | 42 | 34 |
> | **Std. Dev.** | **31.8** | **22.7** | **24.0** |
>
> The sensitivity analysis, as shown in the tables above for the *put bowl & close drawer* task, confirms our choice of hyperparameters. Our selected configuration of $k_o = 10, f_o=20, k_c=2, f_c=2$ consistently yielded high success rates across the diverse missing-modality scenarios (RGB, RGBD, RGBD+All). Crucially, the standard deviation of success rates remains reasonable for variations around our chosen values of $k_o, f_o, \text{and } k_c$, suggesting that the AdaMLI mechanism is relatively robust to small tuning changes for these parameters. However, the standard deviation for the open-to-closed-loop switching parameter, $f_c$, is agreeably high, highlighting its strong influence on the policy's stability and robustness. We have included these detailed sensitivity results in Appendix G of the revised paper to provide the necessary empirical grounding and to serve as a valuable reference for future work adapting this methodology.

---

> ### Author Response · Authors · 2025-11-25
> **Response to Reviewer 62T8 (4/n)**
>
> > Uncontrolled model capacity differences. Because each modality has its own latent vector, the total latent capacity may be larger than in single-latent baselines. The paper should include capacity-controlled comparisons (e.g., matching total latent dimensionality or parameter count) to ensure that improvements are due to the modular design rather than simply increased capacity.
>
> Thank you for raising this important point. We want to ensure that the performance gains demonstrated by NOSTRA are unequivocally attributed to its modular design rather than simply an increase in model capacity. In our experiments, we ensured that for capacity-sensitive comparisons, the baselines were explicitly configured to match the total latent capacity of NOSTRA.
>
> For instance, in Table 2 where we show results using AdaMLI on MimicLabs tasks - which utilize 6 modalities - the BC-NOSTRA architecture employs an 80-dimensional latent state per modality (split into 40 for the stochastic state and 40 for the GRU state). This results in a total consolidated hidden state dimension of 480. We, therefore, configured the single-latent baselines, BC-LSTM and BC-RSSM, with a latent state of size 480 to ensure an equivalent latent capacity across models. We have updated Table 7 in Appendix D to explicitly state this, as well as added another table of hyperparameters for BC-LSTM that shows the latent size for this model highlighting the matching latent dimensionality with our method.
>
> Furthermore, a direct comparison of the total trainable parameters confirms that our model's capacity is comparable to the recurrent baselines:
>
> | Method | Trainable Parameters |
> |---|---|
> | BC-LSTM | 36,705,399 |
> | BC-RSSM | 35,815,399 |
> | BC-NOSTRA | 37,812,103 |
>
> The minor increase in parameters for BC-NOSTRA is primarily due to the per-modality encoders, but this capacity is structurally partitioned, which is the core of our architectural contribution. Since the total parameter count is demonstrably similar to the strong BC-LSTM and BC-RSSM baselines, and their latent state dimensions were explicitly matched for the main multi-modality experiments, the observed performance improvements are directly attributable to the modular per-modality design and the Adaptive Latent Imagination (AdaMLI) mechanism.
>
>
> *We hope that our response addressed your concerns. If so, we would be very grateful if you would consider adjusting your score to reflect the clarifications and additional analysis provided.*
>
>
> References:
>
> [1] Mastering Atari with Discrete World Models, Hafner et al., 2020
>
> [2] Clockwork Variational Autoencoders, Saxena et al., 2021

---

### Official Review · Reviewer_x29x · 2025-11-03

**Soundness:** 3
**Presentation:** 2
**Contribution:** 3
**Rating:** 8
**Confidence:** 4

**Summary:**

In this work, the authors present Nostra: a latent variable model for training imitation learning model from datasets of heterogenous modalities. The problem itself is quite pertinent in the era where large robot models are trained with datasets with many different, somewhat mismatching modalities. The method is justified well, with a structured model that models the seen latents closed loop, and the unseen latents as an open loop dynamics model. Through this prior forcing setup, the model is able to learn from the available modalities, while learning to ignore noisy or not-present modalities. In experiments on simulated tasks, the authors show that the method outperform standard behavior cloning algorithms.

**Strengths:**

1. The algorithm itself is well justified – the training objective shown in equation 2 foilows naturally from the modeling assumptions set up earlier, and it is great to see that with minimal hyperparameter tuning, this objective leads to good objectives for training.
2. On the benchmarks shown in the work, the authors outperform standard behavior cloning algorithms on different manipulation tasks.
3. More impressively, the model is able to robustly predict sensible actions in absence of important modalities, such as visual input.
4. In table 3, the algorithm also shows adaptive behavior when different modalities have different information content.

**Weaknesses:**

1. Spending a bit more time and space towards analyzing and understanding the real robot results would make this work much stronger in practice. Especially in understand how this method performs with longer-horizon uncertainties that come from different dynamics of the robots vs. reality would be interesting to see.
2. The primary evaluations are done on simulated robotic tasks, which while impressive can be less than informative on real robot tasks.

**Questions:**

1. What are the training stability implications of training a model with ELBO on such small amount of data and large amount of modalities (20 demonstrations in the real world)?
2. Currently, the latent transition models are all-to-all, but are there situations where it may be helpful to inject the dependencies as priors?

---

> ### Author Response · Authors · 2025-11-25
> **Response to Reviewer x29x (1/n)**
>
> We thank the reviewer for their time spent in going through our paper, giving insightful comments, and acknowledging that our method is well-justified, robustly predicts sensible actions in the absence of important modalities, and shows adaptive behavior when different modalities have different information content.
>
> We address their concerns below.
>
> > Spending a bit more time and space towards analyzing and understanding the real robot results would make this work much stronger in practice. Especially in understand how this method performs with longer-horizon uncertainties that come from different dynamics of the robots vs. reality would be interesting to see.
>
> We thank the reviewer for their thoughtful comment. We agree that deeper analysis of real-robot results, especially in the context of longer-horizon uncertainties and dynamics mismatch, is highly valuable. However, we believe our real-robot evaluation is comprehensive within the scope of our paper.
>
> We want to emphasize the following:
> 1. Relevance of simulation experiments: The input and output spaces for both our simulated and real-robot experiments are identical, ensuring a direct and relevant transfer of findings from our scalable simulation results to the physical world.
> 2. Complexity of real-robot tasks: We performed experiments on four diverse and challenging real-robot tasks that require full 6-DoF motion and complex behavior for successful completion, notably the *marker in cup* and *pour beans* tasks. On these tasks, we show the usefulness of our methodology (BC-NOSTRA + AdaMLI) for robustness against unseen artifacts in input modalities, such as camera occlusions, and that our policy is deployable with just 30 demonstrations.
> 3. Scope and scalability of experiments: As you noted, our primary evaluations are done in simulated tasks to leverage the achievable scale and prove our contribution with statistical significance. While issues that arise in robot learning due to long-horizon uncertainties are an important challenge, that is not the primary focus of our current paper, and we leave that to future work.
>
> We provide full details about each real-robot task in Appendix H, including 2D visualizations of the learned motions, as well as the initialization ranges of the objects for maximum transparency and reproducibility.
>
> > The primary evaluations are done on simulated robotic tasks, which while impressive can be less than informative on real robot tasks.
>
> While we agree that real-world evaluation is very informative, evaluating our policy for the various capabilities that we wanted to build on a real robot is very time-consuming. Hence, we perform evaluations in simulations that are significantly more scalable. We want to emphasize that the input and output spaces for both our simulated and real-robot experiments are identical, ensuring a direct and relevant transfer of findings. On the real robot, we prove that our policy is deployable with just 30 demonstrations, and we show robustness against camera occlusions. Furthermore, our real-robot evaluation includes four diverse manipulation tasks that require complex behavior for successful completion, demonstrating the practical efficacy of NOSTRA beyond simulation.
>
> > What are the training stability implications of training a model with ELBO on such small amount of data and large amount of modalities (20 demonstrations in the real world)?
>
> We found training models with ELBO (BC-NOSTRA and BC-RSSM) on 30 demonstrations in the real world to be stable. We provide training curves for BC-NOSTRA, BC-RSSM, and Diffusion Policy on [**this link**](https://drive.google.com/file/d/1GT5KvDIH-zVuzX1qC4lwtK_A3tU2NvTw/view?usp=drive_link) for your reference. We find BC-NOSTRA to converge faster than BC-RSSM, highlighting the effects of a modular latent space. Training loss is however not truly representative of real-world performance, and we found that even though BC-RSSM trained stably, during real-world execution, it resulted in jittery and fragile trajectories, while BC-NOSTRA execution was much more stable and surpassed both BC-RSSM and Diffusion Policy in performance (Table 6).

---

> ### Author Response · Authors · 2025-11-25
> **Response to Reviewer x29x (2/n)**
>
> > Currently, the latent transition models are all-to-all, but are there situations where it may be helpful to inject the dependencies as priors?
>
> Thank you for this insightful question. We interpret “all-to-all” as a reference to the fully connected latent transition model where the hidden states of all per-modality latents are consolidated and propagated through the recurrent component (GRU / Modality Transformer) to predict the next prior latent state for all modalities. This approach does not bake in any specific structural dependencies a priori, allowing the model to learn them dynamically from the data.
>
> Regarding whether injecting specific dependencies as priors would be helpful, we agree this is a very interesting direction. Our current design prioritizes maximum flexibility and relies on the learned representations to find the most useful interactions between modalities. However, a structured approach, such as using modality-specific recurrent units (e.g., per-modality GRUs) or a sparse attention mechanism in the transformer, could potentially improve scalability. Exploring structured recurrent models or prior-injected dependencies for the per-modality state consolidation, is a compelling and valuable avenue for future work that could further enhance the model's performance and efficiency. Please let us know in case we misinterpreted your question, and we would appreciate any clarification or further comments or concerns.
>
>
> We hope that our response addressed your concerns. If so, we would be very grateful if you would consider adjusting your score to reflect the clarifications and additional analysis provided.

---

### Official Review · Reviewer_if5D · 2025-11-04

**Soundness:** 2
**Presentation:** 2
**Contribution:** 2
**Rating:** 2
**Confidence:** 3

**Summary:**

The paper presents an approach for robot learning using behavior cloning. The architecture is a state-space model with per-modality latents. The model is evaluated on simulated robotic manipulation tasks.

**Strengths:**

The paper studies an important problem (robot control with multi-modal inputs) and presents an interesting approach (state-space models with modality latents)

**Weaknesses:**

- The motivation for the proposed approach is a bit unclear and would be good to clarify. The paper mentions integrating different modalities, training with subsets of modalities, and noisy modalities which are all related but are not put in perspective.
- Empirical results are overall limited. The approach is evaluated in fairly simple simulated settings that make it a bit hard to judge the importance of the results and how replicable the findings are across different settings.
- The proposed approach contains a number of components and is a bit complex relative to the demonstrated gains over simpler methods. It would be good to discuss the relative tradeoffs further.

**Questions:**

Please see above.

---

> ### Author Response · Authors · 2025-11-25
> **Response to Reviewer if5D (1/n)**
>
> We thank the reviewer for their time and feedback. We respectfully offer the following clarifications on the weaknesses raised, with the intent of illuminating the core contribution and rigor of our submission.
>
> > The motivation for the proposed approach is a bit unclear and would be good to clarify. The paper mentions integrating different modalities, training with subsets of modalities, and noisy modalities which are all related but are not put in perspective.
>
> The primary motivation for NOSTRA is to address the critical, unsolved challenge of multimodal robustness in robot imitation learning - specifically, how to deploy a policy reliably when one or more of its many sensors (modalities) is noisy, occluded, or entirely missing at test time. We draw motivations from humans, who rely on a rich variety of sensing modalities to perform everyday tasks. However, not all modalities are necessary at once: when one becomes unavailable (e.g., vision under occlusion), we draw on experience to bridge the gap, and seamlessly return to the more informative signal once it becomes available again. Hence, achieving human-level robustness in robotic object manipulation requires similar capabilities.
> Drawing from those motivations, we study the following architecture contributions to visuomotor policies: a **modular, per-modality latent space** that allows for **multimodal latent Imagination (MLI)**, a **KL-based adaptive switching mechanism for multimodal robustness (AdaMLI)**, and leveraging the modular latent space when learning from large-scale heterogeneous datasets with non-informative modalities. It is our proposed architectural mechanism that allows the model to selectively switch to its internal world-model prediction for any problematic modality, achieving resilience to missing or noisy inputs. **Reviewer x29x** explicitly acknowledged that the "method is justified well" and that the problem "is quite pertinent" in the current era of large robot models.
>
> > Empirical results are overall limited. The approach is evaluated in fairly simple simulated settings that make it a bit hard to judge the importance of the results and how replicable the findings are across different settings.
>
> We appreciate the desire for even broader validation. However, we believe our empirical evaluation is comprehensive, covering 12 distinct MuJoCo-based manipulation tasks across two benchmarks (MimicGen, MimicLabs), that included multiple input modalities such as multi-view RGB images, depth images, and robot proprioception. Our method consistently demonstrated substantial gains over strong baselines (BC-LSTM, Diffusion Policy, etc.) on being robust against unseen noises, as shown in a total of 16 variations of 4 manipulation tasks in Table 1. To show the efficacy of our proposed AdaMLI mechanism for robustness against unseen noises in multimodal policies, we conducted extensive experiments on a total of 24 variations of 8 manipulation tasks. We showed that our method achieves >20% success on average compared to the next best baseline, and ranks significantly higher than all other methods. Critically, we move beyond simple simulation to include four real-robot tasks on a Franka Panda, where we show robust performance under camera occlusions with only 30 demonstrations.
>
> Moreover, we showed that the multimodal latents in NOSTRA adaptively reduce the information extracted from non-informative inputs (such as occluded view), while increasing focus on other useful inputs. We showed this to hold consistently on 4 MuJoCo-based tasks (Table 3), and also showed that this results in better pre-trained checkpoints for downstream fine-tuning (Table 4) and large-scale co-training (Table 5). We also showed qualitatively how BC-NOSTRA extracts information in its multimodal latents (Figure 5), as well as predicts accurate open-loop actions via multimodal latent imagination (Figure 4).
>
> In summary, we believe the extensive and multi-faceted nature of our empirical evaluation - spanning 12 tasks across two distinct benchmarks, a variety of noisy-modality conditions, and culminating in successful real-robot deployment - convincingly demonstrates the significant contribution and practical efficacy of NOSTRA's modular design and the AdaMLI mechanism in achieving robust, state-of-the-art performance in multimodal robot learning.

---

> ### Author Response · Authors · 2025-11-25
> **Response to Reviewer if5D (2/n)**
>
> > The proposed approach contains a number of components and is a bit complex relative to the demonstrated gains over simpler methods. It would be good to discuss the relative tradeoffs further.
>
>
> We argue that our approach is fairly simple: we present a **joint latent-variable model over actions and observations** because this factorization is what enables the capability we target - **a policy that can act either closed-loop (conditioning on any subset of modalities) or open-loop (with no observations)**. Concretely, we assume the standard state-space generative structure
> $$p_\theta(a_{1:T}, o_{1:T}, z_{0:T}) = p(z_0) \prod_t p_\theta(z_t\mid z_{t-1})\ p_\theta(a_t\mid z_t)\ p_\theta(o_t\mid z_t),$$
> and then train only the action-prediction and latent-dynamics parts, while treating $p_{\theta}(o_t | z_t)$ as an untrained likelihood with fixed variance. Our formulation is no different from a standard VAE, and is not more complex than any existing generative visuomotor policy models such as [1] and [2], or even generative sequence models such as [3] and [4] - albeit novel since variational behavior cloning methods maximize conditional likelihood which results in a policy that requires all inputs at each timestep, unlike ours which can act with or without individual inputs. Finally, the objective in Eq. (2) contains two simple terms: action NLL plus a KL term that regularizes inferred (closed-loop) latents toward the history-conditioned prior (open-loop) latents. Formally, our method is equivalent to maximizing the ELBO of a joint model with an observation decoder whose variance is fixed to $\inf$; the decoder is included to define the posterior family but contributes zero gradient.
>
> Our formulation is necessary to achieve the desired robustness properties (MLI and adaptive execution). The relative tradeoff is therefore the architectural design complexity for a dramatic increase in policy flexibility and real-world robustness. Standard conditional policies, which may appear simpler, fundamentally require all inputs at all times, thereby failing on the core problem we set out to solve. In contrast, our joint model structure and its resulting objective provide a principled way to derive a policy that is inherently robust to missing or noisy inputs.
>
> In this context, we believe the complexity is highly justified by the functional gains. As **Reviewer 62T8** described the approach’s design as “clear and effective” and stated that the “per-modality structure makes the overall approach intuitive and easy to reason about,” demonstrating that the architectural choices, while detailed, are well-motivated and accessible. Furthermore, the significant and consistent empirical performance gains under various noisy-modality conditions, which surpass simpler baselines (BC-LSTM, BC-RSSM, Diffusion Policy), ultimately validate that this apparent complexity is a worthwhile and necessary investment for achieving state-of-the-art multimodal robustness.
>
>
> *We hope that our response addressed your concerns. If so, we would be very grateful if you would consider adjusting your score to reflect the clarifications and additional analysis provided in this rebuttal.*
>
>
> References:
>
> [1] Diffusion Policy: Visuomotor Policy Learning via Action Diffusion, Chi et al., 2023
>
> [2] Implicit Behavioral Cloning, Florence et al., 2021
>
> [3] Learning Latent Dynamics for Planning from Pixels, Hafner et al., 2018
>
> [4] Mastering Atari with Discrete World Models, Hafner et al., 2020

---

### Author Response · Authors · 2025-12-03
**Summary of Rebuttal Response**

We are grateful to the reviewers for their time, valuable feedback, and insightful comments on our submission. We are pleased that the reviewers consistently recognized the core strengths and novelty of our work. Reviewers were uniformly positive about several key aspects of our work:
1. Conceptual Innovation & Design: Our novel approach was praised for its "conceptual innovation" and the "clear and effective design choice of separating the latent representation by modality," making the method “intuitive and easy to reason about.”
2. Robustness and Performance: The model was acknowledged for its ability to "robustly predict sensible actions in absence of important modalities" and for showing "adaptive behavior." The comprehensive experiments were noted for convincingly demonstrating "resilience to RGB, depth, and proprioceptive noise as well as real-robot occlusion."
3. Empirical Rigor: Our experiments were described as "solid and comprehensive," spanning multiple simulated benchmarks, heterogeneous datasets, and successful real-robot evaluations.
## Concerns and Added Analysis

The primary concerns raised across the reviews centered on the theoretical justification of the training objective and the empirical validation of the proposed switching mechanism. We have addressed these with detailed clarifications and added new quantitative analysis.

1. **Inconsistency between Joint Generative Model and ELBO Objective** *(Reviewer 62T8)* - We provided a detailed technical clarification that our objective is still the ELBO of the joint model. This is achieved by fixing the observation decoder's variance to infinity, which removes the reconstruction term without changing the underlying variational formulation. This structure is fundamentally necessary to enable the open-loop "Multimodal Latent Imagination (MLI)" capability.
2. **Justification for Stop-Gradient KL Mixture** *(Reviewer 62T8, z4L7)* - We clarified that the stop-gradient mixture, $\gamma KL(q | stopgradient(p)) + (1 - \gamma) KL(stopgradient(q) | p)$, has the same value as the standard KL term and thus preserves the ELBO. Its purpose is to tune the gradient flow. **New quantitative analysis** on the *coffee D2* task demonstrated that $\gamma=0.1$ improves action reconstruction error when using the learned prior (open-loop latents), validating its role in strengthening the MLI mechanism.
3. **Sensitivity Analysis of AdaMLI Hyperparameters** *(Reviewer 62T8, z4L7)* - We presented **new quantitative analysis** on the sensitivity of model performance on AdaMLI parameters ($k_o, f_o, k_c, f_c$) for the *put bowl & close drawer* task, confirming our choice of hyperparameters. The mechanism was found to be relatively robust to small changes in $k_o, f_o,$ and $k_c$, as shown by their reasonable standard deviations. However, the open-to-closed-loop switching parameter $f_c$ showed a high standard deviation, indicating its strong and sensitive influence on the policy's stability and robustness. These detailed results were included in Appendix G for empirical grounding and future reference.
4. **Unclear Motivation/Empirical Scope** *(Reviewer if5D, x29x)* - We clarified the primary motivation as addressing the **critical, unsolved challenge of multimodal robustness** when sensors are noisy or missing at test time. We emphasized that our comprehensive evaluation, spanning 12 tasks across two benchmarks and four real-robot tasks, already demonstrates the significant contribution and practical efficacy of the approach.
5. **Training Stability on Small Data** *(Reviewer x29x)* - We confirmed that training with the ELBO objective was stable even with limited real-world data (30 demonstrations). We provided a link to the **training curves** for reference, which also showed that our modular BC-NOSTRA model converges faster than the BC-RSSM baseline.

We believe that the clarifications around theoretical formulation, and the added quantitative analyses on gradient flow and hyperparameter sensitivity comprehensively address the theoretical and empirical concerns of the reviewers. We hope this additional context sheds further light on our contributions and clarifies any remaining concerns.

---

### Meta-Review · Area_Chair_KRym · 2026-01-07

**Summary:**

In the initial review, reviewers' concerns include unclear illustration, over reliance on simulation, and very simple real world experiments.

After rebuttal, I think the concerns regarding illustration of methods are clarified.

However, considering the high stanard of ICLR and the competitive works in the field, I recommand rejection since weak real world experiments significantly weakens the soundness of robotics research.

The authors are encouraged to submit to a new venue with improved real world experiments.

**Reviewer Concerns:**

In the initial review, reviewers' concerns include unclear illustration, over reliance on simulation, and very simple real world experiments.

After rebuttal, I think the concerns regarding illustration of methods are clarified.

However, I do not think the concerns regarding real world robot experiments are fully resolved.

**Reviewer Scores:**

Reviewer if5D and 62T8 might keep their negative scores (if5D might raise to 4 as the unclear details are clarified) while Reviewer z4L7 and x29x might keep their positive score.

---

### Decision · Program_Chairs · 2026-01-26

Reject